# Latent Hierarchical Causal Structure Discovery with Rank Constraints

**Biwei Huang** [*1] **Charles Low** [*1], **Feng Xie**[3], **Clark Glymour**[1], **Kun Zhang**[1,2]
[1] Carnegie Mellon University
[2] Mohamed bin Zayed University of Artificial Intelligence
[3] Beijing Technology and Business University, China
{bwei.huang, charleslow88, xiefeng009}@gmail.com,
cg09@andrew.cmu.edu, kunz1@cmu.edu

## Abstract

Most causal discovery procedures assume that there are no latent confounders in the system, which is often violated in real-world problems. In this paper, we consider a challenging scenario for causal structure identification, where some variables are latent and they form a hierarchical graph structure to generate the measured variables; the children of latent variables may still be latent and only leaf nodes are measured, and moreover, there can be multiple paths between every pair of variables (i.e., it is beyond tree structure). We propose an estimation procedure that can efficiently locate latent variables, determine their cardinalities, and identify the latent hierarchical structure, by leveraging rank deficiency constraints over the measured variables. We show that the proposed algorithm can find the correct Markov equivalence class of the whole graph asymptotically under proper restrictions on the graph structure.

## 1 Introduction

In many cases, the common assumption in causal discovery algorithms–no latent confounders–may not hold. For example, in complex systems, it is usually hard to enumerate and measure all task-related variables, so there may exist latent variables that influence multiple measured variables, the ignorance of which may introduce spurious correlations among measured variables. Much effort has been made to handle the confounding problem in causal structure learning. One research line considers the causal structure over measured variables, including FCI and its variants [Spirtes et al., 2000, Pearl, 2000, Colombo et al., 2012, Akbari et al., 2021], matrix decomposition-based approaches [Chandrasekaran et al., 2011, 2012, Frot et al., 2019], and over-complete ICA-based ones [Hoyer et al., 2008, Salehkaleybar et al., 2020].

Another line focuses on identifying the causal structure among latent variables, including Tetrad condition-based approaches [Silva et al., 2006, Kummerfeld and Ramsey, 2016], high-order moments-based ones [Shimizu et al., 2009, Cai et al., 2019, Xie et al., 2020, Adams et al., 2021, Chen et al., 2022], matrix decomposition-based approach Anandkumar et al. [2013], copula model-based approach [Cui et al., 2018], mixture oracles-based approach [Kivva et al., 2021], and multiple domains-based approach [Zeng et al., 2021]. Moreover, regarding the scenario of latent hierarchical structures, previous work along this line assumes a tree structure [Pearl, 1988, Zhang, 2004, Choi et al., 2011, Drton et al., 2017], where there is one and only one undirected path between every pair of variables. This assumption is rather restrictive and the structure in real-world problems could be more complex–beyond a tree.

In this paper, we consider a more challenging scenario where latent causal variables form a hierarchical graph structure to generate measured variables—the children of latent variables may still be latent

---

[*]These authors contributed equally to this work.

36th Conference on Neural Information Processing Systems (NeurIPS 2022).

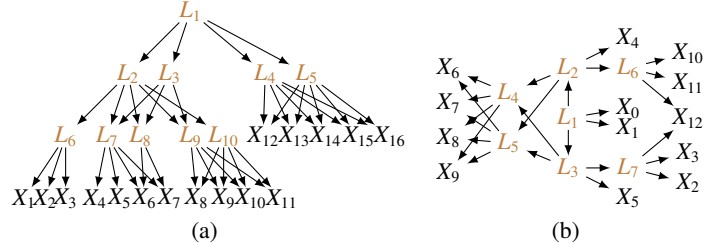

| Pa: | Parents |
|---|---|
| PCh: | Pure children |
| PDe: | Pure descendants |
| Gp: | Grandparents |
| Sib: | Siblings |
| $\mathcal{M}$: | Measured pure descendants |

Table 1: Graphical notations.

Figure 1: Example hierarchical graphs that our method can handle, where $X_i$ are measured variables and $L_i$ are latent variables.

and only the leaf nodes are measured, and moreover, there can be multiple paths between every pair of variables (see the example hierarchical graphs in Figure 1). We aim to find out identifiability conditions of the hierarchical structure that are as mild as possible, and meanwhile, develop an efficient algorithm with theoretical guarantees to answer the following questions. (1) How can we locate latent parents for both measured and latent variables, as well as determining the cardinality of the latent parents, by only providing the leaf nodes? (2) How can we identify the causal relationships among latent variables and those from latent variables to measured variables?

Interestingly, we can answer these questions by properly making use of rank deficiency constraints; finding and leveraging rank properties in specific ways enable us to identify the Markov equivalence class of the whole graph, under appropriate conditions. Our contribution is mainly two-fold:

- We propose a structure identification algorithm that can efficiently locate latent variables (including their cardinalities) and identify the latent hierarchical structure, by leveraging the rank deficiency.
- We show that the proposed algorithm can find the correct graph asymptotically under mild restrictions of the graph structure. Roughly speaking, we show that it is sufficient to have $k + 1$ pure children (which can be latent), as well as another $k + 1$ neighbors, to identify the latent variable set with size $k$ (see the detailed conditions in Definition 4 and Condition 1).

It is worth mentioning that rank constraints have been used in previous methods [Silva et al., 2006, Kummerfeld and Ramsey, 2016], but they assume that each latent variable has three measured ones as children and each measured variable has only one latent parent. There are also other methods for latent structure learning; for instance, Anandkumar et al. [2013], which uses matrix decomposition, needs $3k$ measured children, and the GIN-based method [Xie et al., 2020], which makes use of high-order statistics, needs $2k$ measured children. However, all those developments require that every latent variable should have measured variables as children. Very recently, Xie et al. [2022] proposes an approach for latent hierarchical structure by leveraging the GIN condition under linear non-Gaussian models, but it assumes that each variable has only one parent, where both figures in Figure 1 do not satisfy.

This paper is organized as follows. In Section 2, we give formal definitions of the latent hierarchical causal model under investigation and give conditions that are essential to the identifiability of the graph structure. In Section 3, we propose an efficient algorithm that makes use of rank-deficiency constraints to identify the latent hierarchical structure. Moreover, we show theoretically in Section 4 that the proposed algorithm outputs the correct Markov equivalence class of the whole graph asymptotically. In Section 5, we empirically validate the proposed approach on synthetic data. Notations for graphical representations that are used in the paper are provided in Table 1.

## 2 Latent Hierarchical Causal Model

In this paper, we focus on latent hierarchical causal model with graph structure $\mathcal{G}$, where both measured variables $\mathbf{X}_{\mathcal{G}} = \{X_1, \cdots, X_m\}$ and latent variables $\mathbf{L}_{\mathcal{G}} = \{L_1, \cdots, L_n\}$ are generated by their latent parents in a directed acyclic graph (DAG) with linear relationships:

$$X_i = \sum_{L_j \in Pa(X_i)} b_{ij} L_j + \varepsilon_{X_i}, \quad L_j = \sum_{L_k \in Pa(L_j)} c_{jk} L_k + \varepsilon_{L_j}, \quad (1)$$

where $b_{ij}$ and $c_{jk}$ are the causal strength from $L_j$ to $X_i$ and from $L_k$ to $L_j$, respectively, and $\varepsilon_{X_i}$ and $\varepsilon_{L_j}$ are noise terms that are independent of each other. Without loss of generality, we assume that all variables in $\mathbf{X}_{\mathcal{G}}$ and $\mathbf{L}_{\mathcal{G}}$ have zero mean.

Below, we first give the general definition of a linear latent hierarchical graphical model in Definition 1. Then we give more detailed conditions on the graph structure (Definitions 2-4) that are essential to formalize the identifiability condition of the latent hierarchical structure.

**Definition 1** (Linear Latent Hierarchical (L²H) Model). *A graphical model, with its graph $\mathcal{G} = (\mathbf{V}_{\mathcal{G}}, \mathbf{E}_{\mathcal{G}})$, is a linear latent hierarchical model if:*

1. *$\mathbf{V}_{\mathcal{G}} = \mathbf{X}_{\mathcal{G}} \bigcup \mathbf{L}_{\mathcal{G}}$, where $\mathbf{X}_{\mathcal{G}}$ is the set of measured variables and $\mathbf{L}_{\mathcal{G}}$ is the set of latent variables,*

2. *there is at least one undirected path between every pair of variables, and*

3. *each variable in $\mathbf{X}_{\mathcal{G}}$ and $\mathbf{L}_{\mathcal{G}}$ are generated by the structural equation models in Eq. 1.*

Generally, without further constraints, the causal structure of the L²H model is hard to be identified. It has been shown that if the underlying graph structure satisfies a tree [Pearl, 1988], then the structure is identifiable. However, this structural constraint may be too strong to hold in many real-world problems. In this paper, we give sufficient conditions that are much milder than previous ones, as well as an efficient search algorithm, for the identifiability of the causal structure.

We now give the corresponding definitions, including *pure children*, *pure descendants*, and *effective cardinality*, that will be used in the identifiability condition, together with illustrative examples.

**Definition 2** (Pure Children). *Variables $\mathbf{V}$ are pure children of a set of latent variables $\mathbf{L}$ in a graph $\mathcal{G}$, if $Pa_{\mathcal{G}}(\mathbf{V}) = \mathbf{L}$ and $\mathbf{V} \cap \mathbf{L} = \emptyset$. That is, $\mathbf{V}$ have no other parents than $\mathbf{L}$. We denote the pure children of $\mathbf{L}$ by $PCh_{\mathcal{G}}(\mathbf{L})$.*

Accordingly, *Pure Descendants* of a set of latent variables $\mathbf{L}$ are defined as all recursive pure children of $\mathbf{L}$ (including $PCh_{\mathcal{G}}(\mathbf{L}), PCh_{\mathcal{G}}(PCh_{\mathcal{G}}(\mathbf{L}))$, etc.), denoted by $PDe_{\mathcal{G}}(\mathbf{L})$. Furthermore, measured variables that are pure descendent of $\mathbf{L}$ are called *Measured Pure Descendants*, denoted by $\mathcal{M}_{\mathcal{G}}(\mathbf{L})$.

**Example 1.** *In Figure 1(a), the pure children of $\{L_2, L_3\}$ are $\{L_6, \cdots, L_{10}\}$, its pure descendants are $\{L_6, \cdots, L_{10}, X_1, \cdots, X_{11}\}$, and its measured pure descendants are $\{X_1, \cdots, X_{11}\}$.*

**Definition 3** (Effective Cardinality). *For a set of latent variables $\mathbf{L}$, denote by $\mathbf{C}$ the largest subset of $PCh_{\mathcal{G}}(\mathbf{L})$ such that there is no subset $\mathbf{C}' \subseteq \mathbf{C}$ satisfying $|\mathbf{C}'| > |Pa_{\mathcal{G}}(\mathbf{C}')|$ and $|Pa_{\mathcal{G}}(\mathbf{C}')| < |\mathbf{L}|$. Then, the effective cardinality of $\mathbf{L}$'s pure children is $|\mathbf{C}|$.*

In the case when $\mathbf{L}$ and its pure children are fully connected, the effective cardinality is just the cardinality of the pure children of $\mathbf{L}$. However, for Figure 1(a), the effective cardinality of the pure children of $\{L_7, L_8\}$ is 3, because here the largest subset that satisfies the condition is $\{X_5, X_6, X_7\}$.

We further define *latent atomic cover* that constrains the number of pure children and neighbours for latent variables, which are essential for structural identifiability.

**Definition 4** (Latent Atomic Cover). *Let $\mathbf{L} = \{L_1, ..., L_k\}$ be a set of latent variables in graph $\mathcal{G}$, with $|\mathbf{L}| = k$. We say that $\mathbf{L}$ is a latent atomic cover if the following conditions are met:*

1. *there exists a subset of pure children $\mathbf{C}' \subseteq PCh_{\mathcal{G}}(\mathbf{L})$ with effective cardinality $\geq k + 1$;*

2. *there exists a neighbour set $\mathbf{B}$ to $\mathbf{L}$ s.t. $\mathbf{B} \cap \mathbf{C}' = \emptyset$ and $|\mathbf{B}| = k + 1$;*

3. *there does not exist a partition of $\mathbf{L} = \mathbf{L}_1 \cup \mathbf{L}_2$, so that both $\mathbf{L}_1, \mathbf{L}_2$ satisfy conditions 1 and 2 and $\{PCh_{\mathcal{G}}(\mathbf{L}_1) \cup PCh_{\mathcal{G}}(\mathbf{L}_2)\} \backslash \mathbf{L} = PCh_{\mathcal{G}}(\mathbf{L})$.*

**Example 2.** *In Figure 1(a), $\mathbf{L} = \{L_2, L_3\}$ is a latent atomic cover with $k = 2$, because (1) there exists a subset of pure children $\mathbf{C}' = \{L_6, L_7, L_8\}$ with effective cardinality $3 = k + 1$, (2) there exists a neighbor set $\mathbf{B} = \{L_1, L_9, L_{10}\}$, s.t. $\mathbf{B} \cap \mathbf{C}' = \emptyset$ and $|\mathbf{B}| = 3 = k + 1$, and (3) neither $\{L_2\}$ or $\{L_3\}$ satisfies the above two conditions.*

We now give the conditions for structural identifiability from measured variables $\mathbf{X}_{\mathcal{G}}$ alone, including those on the structural constraints (Condition 1) and rank faithfulness assumption (Condition 2).

**Condition 1** (Irreducible Linear Latent Hierarchical (IL²H) Graph). *An L²H graph $\mathcal{G}$ is an IL²H graph if*

1. *every latent variable $L \in \mathbf{L}$ in $\mathcal{G}$ belongs to at least one latent atomic cover,*

2. *for any pair of latent atomic covers $(\mathbf{L}_A, \mathbf{L}_B)$, if $PDe_{\mathcal{G}}(\mathbf{L}_A) \bigcap PDe_{\mathcal{G}}(\mathbf{L}_B) \neq \emptyset$, then either (a) $\mathbf{L}_A \subset \mathbf{L}_B$ or (b) $\mathbf{L}_A \subset PDe_{\mathcal{G}}(\mathbf{L}_B)$ or (c) $\mathbf{L}_B \subset \mathbf{L}_A$ or (d) $\mathbf{L}_B \subset PDe_{\mathcal{G}}(\mathbf{L}_A)$, and*

3. *for any three latent atomic covers $\mathbf{L}_A, \mathbf{L}_B, \mathbf{L}_C$, if the causal structure satisfies $\mathbf{L}_A \to \mathbf{L}_B \to \mathbf{L}_C$ or $\mathbf{L}_A \gets \mathbf{L}_B \to \mathbf{L}_C$, and $|\mathbf{L}_B| = k$, then $\mathbf{L}_B$ has $2k$ neighbors, except for $\mathbf{L}_A, \mathbf{L}_C$ and the parents in the v structure where $\mathbf{L}_B$ is a collider.*

Next, we give the faithfulness assumption, which holds for generic covariance matrices consistent with $\mathcal{G}$ [Spirtes, 2013].

**Condition 2** (Rank Faithfulness). *A probability distribution P is rank faithful to a DAG $\mathcal{G}$ if every rank constraint on a sub-covariance matrix that holds in P is entailed by every linear structural model with respect to $\mathcal{G}$.*

We will show in Section 4 that if the underlying latent hierarchical graph satisfies an IL$^2$H graph and the rank faithfulness holds, then the location and cardinality of latent variables, and the causal structure among latent atomic covers and that from latent atomic covers to measured variables, are identifiable with appropriate search procedures. Below, let us first present the identification procedure.

## 3   Structure Identification with Rank-Deficiency Constraints

We propose an identification algorithm (Algorithm 1) to identify the structure of IL$^2$H graphs, by leveraging rank-deficiency constraints of measured variables. In particular, the algorithm includes three phases: finding causal clusters and assigning latent atomic covers in a greedy manner ("*find-CausalClusters*"), refining incorrect clusters and covers due to the greedy search ("*refineClusters*"), and refining edges and finding v structures ("*refineEdges*").

---
**Algorithm 1:** Latent Hierarchical Causal Structure Discovery

**Input**   :Date from a set of measured variables $\mathbf{X}_{\mathcal{G}}$
**Output**:Markov equivalence class $\mathcal{G}'$
1 $\mathcal{G}'$ = findCausalClusters ($\mathbf{X}_{\mathcal{G}}$) ;          // find clusters and assign latent covers greedily
2 $\mathcal{G}'$ = refineClusters ($\mathcal{G}'$) ;    // refine incorrect clusters and covers from greedy search
3 $\mathcal{G}'$ = refineEdges ($\mathcal{G}'$) ;                      // refine some edges and find v structures

---

Before describing the identification algorithm, we first give a theorem that relates the graphical structure of an IL$^2$H graph to the rank constraints over the covariance matrix of measured variables.

**Theorem 1** (Graphical Implication of Rank Constraints in IL$^2$H Graphs). *Suppose $\mathcal{G}$ satisfies an IL$^2$H graph. Under the rank faithfulness assumption, the cross-covariance matrix $\Sigma_{\mathbf{X}_A, \mathbf{X}_B}$ over measured variables $\mathbf{X}_A$ and $\mathbf{X}_B$ in $\mathcal{G}$ (with $|\mathbf{X}_A|, |\mathbf{X}_B| > r$) has rank r, if and only if there exists a subset of latent variables $\mathbf{L}$ with $|\mathbf{L}| = r$ such that $\mathbf{L}$ d-separates $\mathbf{X}_A$ from $\mathbf{X}_B$, and there is no $\mathbf{L}'$ with $|\mathbf{L}'| < |\mathbf{L}|$ that d-separates $\mathbf{X}_A$ from $\mathbf{X}_B$. That is,*

$$rank(\Sigma_{\mathbf{X}_A, \mathbf{X}_B}) = min\{|\mathbf{L}| : \mathbf{L} \text{ d-separates } \mathbf{X}_A \text{ from } \mathbf{X}_B\}.$$

For instance, in Figure 1(a), $rank(\Sigma_{\{X_1, X_2\}, \{X_3, X_4\}}) = 1$, because $L_6$ d-separates $\{X_1, X_2\}$ from $\{X_3, X_4\}$ with $|L_6| = 1$.

**Rank Test.**   We test rank deficiency by leveraging canonical correlations [Anderson, 1984]. Specifically, the number of non-zero canonical correlations between two random vectors is equal to the rank. Denote by $\alpha_i$ the $i$-th canonical correlation coefficient between $\mathbf{X}_A$ and $\mathbf{X}_B$. Then under the null hypothesis that $\mathtt{rank}(\Sigma_{\mathbf{X}_A, \mathbf{X}_B}) \leq r$, the statistics $-(N - (p + q + 3)/2) \sum_{i=r+1}^{\max(p,q)} \log(1 - \alpha_i^2)$ is approximately $\chi^2$ distributed with $(p - r)(q - r)$ degrees of freedom, where $p = |\mathbf{X}_A|$, $q = |\mathbf{X}_B|$, and $N$ is the sample size.

We further show that for any subset of latent variables in an IL$^2$H graph, we can use the measured variables as surrogates to estimate the rank, as indicated in the following theorem.

**Theorem 2** (Measurement as a surrogate). *Suppose $\mathcal{G}$ is an IL$^2$H graph. Denote by $\mathbf{A}, \mathbf{B} \subseteq \mathbf{V}_{\mathcal{G}}$ two subsets of variables in $\mathcal{G}$, with $\mathbf{A} \cap \mathbf{B} = \emptyset$. Furthermore, denote by $\mathbf{X}_A$ the set of measured variables that are d-separated by $\mathbf{A}$ from all other measures, and by $\mathbf{X}_B$ the set of measured variables that are d-separated by $\mathbf{B}$ from all other measures. Then $\mathtt{rank}(\Sigma_{\mathbf{A}, \mathbf{B}}) = \mathtt{rank}(\Sigma_{\mathbf{X}_A, \mathbf{X}_B})$.*

A special case when Theorem 2 holds is that $\mathbf{X}_A$ and $\mathbf{X}_B$ are the measured pure descendants of $\mathbf{A}$ and $\mathbf{B}$, respectively. Note that Theorem 1 is a special case of Theorem 2.8 in Sullivant et al. [2010] when applied to IL$^2$H graphs. Different from the setting in Sullivant et al. [2010] where access to the full covariance matrix $\Sigma_{\mathbf{V}_{\mathcal{G}}, \mathbf{V}_{\mathcal{G}}}$ is assumed, we only have access to the covariance matrix $\Sigma_{\mathbf{X}_{\mathcal{G}}, \mathbf{X}_{\mathcal{G}}}$ over the measured variables $\mathbf{X}_{\mathcal{G}}$, which we will use to infer the causal structure over the entire graph $\mathcal{G}$. For this reason, although we can infer the number of latent variables that d-separate any two sets of measured variables $\mathbf{X}_A, \mathbf{X}_B$, we cannot directly know the exact location of these variables in the graph. Fortunately, the structure constraints of the IL$^2$H graph and Theorem 2 will allow us to reconstruct the graph with certain search procedures, as shown below.

## 3.1 Phase I: Finding Causal Clusters

We start to discover clusters in a recursive and greedy manner, by performing rank-deficiency tests over measured variables. We denote by $\mathcal{S}$ a set of active variables that is under investigation; $\mathcal{S}$ is set to $\mathbf{X}_{\mathcal{G}}$ initially and will be updated to include latent atomic covers when rank deficiency is discovered. Below, we first give the definition about set size and the definition of *atomic rank-deficiency set*, which will be used in Rule 1 and Algorithm 2.

**Definition 5** (Set Size). *Suppose $\mathcal{S}$ is a set. We denote by $|\mathcal{S}|$ the cardinality of $\mathcal{S}$; that is, $|\mathcal{S}|$ is the number of elements $S_i$ in $\mathcal{S}$, for $S_i \in \mathcal{S}$. We denote by $\|\mathcal{S}\|$ the number of variables in $\mathcal{S}$, where $\|\mathcal{S}\| = |\bigcup_i S_i|$, for $S_i \in \mathcal{S}$.*

For example, suppose $\mathcal{S} = \{\{L_1\}, \{L_6\}, \{L_7, L_8\}, \{L_9, L_{10}\}\}$. Then $|\mathcal{S}| = 4$ and $\|\mathcal{S}\| = 6$.

**Definition 6** (Atomic Rank-Deficiency Set). *Given a graph $\mathcal{G}'$. Denote by $\mathcal{S} \subseteq V_{\mathcal{G}'}$ an active set of variables that is under investigation. Let $\mathbf{A} \subset \mathcal{S}$ and $\mathbf{B} = \mathcal{S} \backslash \mathbf{A}$. Denote by $\mathbf{X}_A = \mathcal{M}_{\mathcal{G}'}(\mathbf{A})$ and $\mathbf{X}_B = \mathcal{M}_{\mathcal{G}'}(\mathbf{B})$ the measured pure descendants of $\mathbf{A}$ and $\mathbf{B}$, respectively. If (1) $\Sigma_{\mathbf{A},\mathbf{B}}$ is rank deficient, i.e., $\mathrm{rank}(\Sigma_{\mathbf{A},\mathbf{B}}) < \min\{\|\mathbf{A}\|, \|\mathbf{B}\|\}$), and (2) no proper subset of $\mathbf{A}$ is rank deficient, then $\mathbf{A}$ is called an atomic rank-deficient set, and $\Sigma_{\mathbf{A},\mathbf{B}}$ can be estimated by $\Sigma_{\mathbf{X}_A,\mathbf{X}_B}$.*

For example, in Figure 1(a), suppose now we have the measured variables $\mathbf{X}_{\mathcal{G}}$ under investigation, so the active variable set $\mathcal{S} = \mathbf{X}_{\mathcal{G}}$. $\mathbf{A} = \{X_1, X_2\} \subset \mathcal{S}$ forms an atomic rank-deficiency set because the cross-covariance matrix of $\mathbf{X}_A$ against all other measures has rank 1 (i.e., rank deficient). According to Theorem 1, the rank deficiency occurs because the latent parent $L_6$ of $\{X_1, X_2\}$ d-separates them from all other measures. This naturally leads to the following rule that assigns a latent atomic cover over the rank-deficiency set:

> **Rule 1:** If $\mathbf{A}$ is a rank-deficiency set with $\mathrm{rank}(\Sigma_{\mathbf{X}_A,\mathbf{X}_B}) = k$, then assign a latent atomic cover $\mathbf{L}$ of size $k$ as the parent of every variable $A_i \in \mathbf{A}$.

Later in Phase II, we will show that Rule 1 may not correctly identify the latent atomic cover in some cases, and, accordingly, we will further provide an efficient revision procedure in Phase II. At the current phase, we use Rule 1, together with certain search procedures, to identify the clusters and latent atomic covers. The detailed search procedure of Phase I is given in Algorithm 2 (*findCausalClusters*), which tests for rank deficiency recursively to discover clusters of variables and their latent atomic clusters. The set of active variables $\mathcal{S}$ is set to $\mathbf{X}_{\mathcal{G}}$ initially (line 1) and will be updated as the search goes on (line 14). We start to identify the latent atomic cover with size $k = 1$ (line 1). We consider any subset of the latent atomic covers in $\mathcal{S}$ and replace them with their pure children, resulting in $\tilde{\mathcal{S}}$ (line 4). Then we draw a subset of variables $\mathbf{A} \subset \tilde{\mathcal{S}}$ with cardinality at least $k + 1$ and conduct a rank deficiency test of $\mathbf{A}$ against $\mathcal{S} \backslash \mathbf{A}$ by estimating the rank of $\mathrm{rank}(\Sigma_{\mathbf{X}_A,\mathbf{X}_B})$ (lines 6-7). Note that here $\mathbf{X}_A$ and $\mathbf{X}_B$ are the measured pure descendants of $\mathbf{A}$ and $\mathbf{B}$, respectively, in the currently learned graph. This step is repeated until all subsets are tested (lines 5-9). If rank deficiency is found, we merge the overlapping groups into a cluster and add latent covers over the cluster (lines 10-14). We further reset $k = 1$ and resume the search (line 15). Otherwise, if no latent cover is found, we increment $k = k + 1$ (line 17). This procedure is repeated until no more clusters are found. Finally, we connect the elements in $\mathcal{S}$ into a chain structure (line 19). Figure 2 illustrates an example procedure of finding new clusters by applying *findCausalClusters* to the measured variables $\mathbf{X}_{\mathcal{G}}$ generated from the structure in Figure 1(a).

## 3.2 Phase II: Refining Clusters

As we have mentioned above, the naive assignment of causal clusters in Algorithm 2 (*findCausalClusters*) may not be correct in some cases. In this section, we provide a precise characterization of the cases where *findCausalClusters* incorrectly clusters variables and, accordingly, propose an efficient algorithm to correct such cases.

Take Figure 3(a) as an example to illustrate the issue. By applying *findCausalClusters* to the measured variables $\mathbf{X}_{\mathcal{G}}$, at $k = 2$, we discovered that $\{X_9, X_{10}, X_{11}\}$ form a cluster and then set a 2-latent atomic cover $\{L'_1, L'_2\}$ over it (Figure 3(b)). This is not correct, because $X_9, X_{10}, X_{11}$ actually belong to three different 1-latent atomic covers, $L_1, L_2, L_3$, respectively. The incorrect clustering and covering are because when discovering the rank-deficiency set $\{X_9, X_{10}, X_{11}\}$, the latent covers $\{L_4, L_5\}, \{L_6, L_7\}$ have not been identified yet. If $\{L_4, L_5\}$ had already been discovered, we would correctly find that $\{X_9, L_4, L_5\}$ form a 1-latent atomic cover, and the algorithm would proceed correctly thereafter. More generally, *findCausalClusters* may set incorrect cluster for the rank-deficiency set $\{X_9, X_{10}, X_{11}\}$, in

**Algorithm 2:** Phase I: findCausalClusters

**Input**  : Data from a set of measured variables $\mathbf{X}_\mathcal{G}$
**Output**: Graph $\mathcal{G}'$

1  Active set $\mathcal{S} \leftarrow \mathbf{X}_\mathcal{G}$; $k \leftarrow 1$;
2  **repeat**
3      **repeat**
4          draw a set of latent atomic covers $\tilde{\mathbf{L}} \subset \mathcal{S}$; let $\tilde{\mathcal{S}} = (\mathcal{S}\backslash\tilde{\mathbf{L}}) \cup (\cup_{L_i \in \tilde{\mathbf{L}}} PCh(L_i))$;
5          **repeat**
6              draw a set of test variables $\mathbf{A} \subset \tilde{\mathcal{S}}$, with $\|\mathbf{A}\| \geq k + 1$, and let $\mathbf{B} \leftarrow \tilde{\mathcal{S}}\backslash\mathbf{A}$;
7              $k' \leftarrow \texttt{rank}(\Sigma_{\mathbf{X}_A, \mathbf{X}_B})$ estimated by rank deficiency test;
8              **if** $k' < k + 1$ **then**  rank deficiency found and keep track of this set $\mathbf{A}$ ;
9          **until** *all subsets* $\mathbf{A}$ *exhausted*;
10          **if** *any groups with rank deficiency are found* **then**
11              merge overlapping clusters; identify lowest rank $k'$ found;
12              **foreach** *discovered cluster of variables* $\mathbf{A}$ *with rank* $k'$ **do**
13                  create latent cover $\mathbf{L}$ with cardinality $k'$ as parents of $\mathbf{A}$;
14                  $\mathcal{S} = (\mathcal{S}\backslash\mathbf{A}) \cup \mathbf{L}$;
15              $k = 1$; **break**;
16      **until** *all subsets* $\tilde{\mathbf{L}}$ *exhausted*;
17      **if** *no group with rank deficiency is found* **then**  $k = k + 1$;
18  **until** *no more clusters are found*;
19  for all $S_i \in \mathcal{S}$, connect them into a chain structure;
20  **return** $\mathcal{G}'$

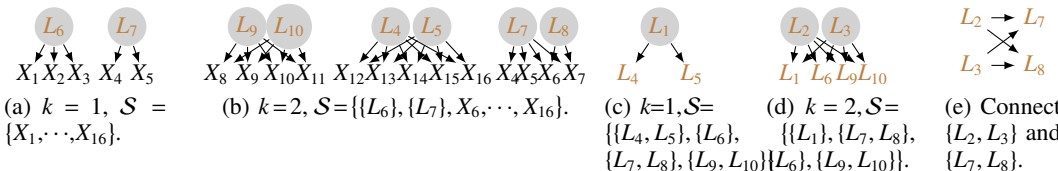

(a) $k = 1$, $\mathcal{S} =$ $\{X_1, \cdots, X_{16}\}$.

(b) $k=2$, $\mathcal{S} = \{\{L_6\}, \{L_7\}, X_6, \cdots, X_{16}\}$.

(c) $k=1$, $\mathcal{S} =$ $\{\{L_4, L_5\}, \{L_6\},$ $\{L_7, L_8\}, \{L_9, L_{10}\}\}$

(d) $k = 2$, $\mathcal{S} =$ $\{\{L_1\}, \{L_7, L_8\},$ $L_6\}, \{L_9, L_{10}\}\}$.

(e) Connect $\{L_2, L_3\}$ and $\{L_7, L_8\}$.

Figure 2: An illustration of Algorithm 2 that discovers new clusters in sequence (marked with gray circle) by applying *findCausalClusters* to the measured variables $\mathbf{X}_\mathcal{G}$ generated from the structure in Figure 1(a). Specifically, we first set $k = 1$ and the active set is $\mathcal{S} = \{X_1, \cdots, X_{16}\}$ and $\tilde{\mathcal{S}} = \mathcal{S}$, and we can find the clusters in (a), and no further cluster can be found with $k = 1$. Then we increase $k$ to 2 with the active set $\mathcal{S} = \{\{L_6\}, \{L_7\}, X_6, \cdots, X_{16}\}$ and $\tilde{\mathcal{S}} = \mathcal{S}$, and then we can find the clusters in (b). Then, the active set is $\mathcal{S} = \{\{L_4, L_5\}, \{L_6\}, \{L_7, L_8\}, \{L_9, L_{10}\}\}$ and we set back $k = 1$, and when $\tilde{\mathcal{S}} = \{\{L_4, L_5\}, X_1, \cdots, X_{11}\}$ we find the cluster in (c). Note that when testing the rank over $\{L_4, L_5\}$ against other variables, we use their measured pure descendants in the currently estimated graph instead. The above procedure is repeated to further find the cluster in (d). Finally, when there are no enough variables for testing, we connect the elements in the active variable set: connecting $\{L_2, L_3\}$ to $\{L_7, L_8\}$ in (e).

the case when their parents $L_1, L_2, L_3$ split the graph into two or more disconnected graphs. This result is formally stated in the following definition and theorem.

**Definition 7** (Bond Set). *Consider a set of measured variables* $\mathbf{X} \subseteq \mathbf{X}_\mathcal{G}$, *and a minimal set of latent variables* $\mathbf{L} \subseteq \mathbf{L}_\mathcal{G}$ *that d-separate* $\mathbf{X}$ *from all other measures* $\mathbf{X}' := \mathbf{X}_\mathcal{G}\backslash\mathbf{X}$. *We say that* $\mathbf{X}$ *is a bond set if* $\mathbf{L}$ *also d-separates some partition* $\mathbf{X}_A \subset \mathbf{X}', \mathbf{X}_B \subset \mathbf{X}'$ *from one another.*

From the previous example, we have seen that with the existence of bond sets, *findCausalClusters* may end up with incorrect clusters and covers. The following theorem shows that the presence of bond sets is the only reason for incorrect clusters and covers with *findCausalClusters*.

**Theorem 3** (Correct Cluster Condition). *Suppose* $\mathcal{G}$ *is an* $IL^2H$ *graph with measured variables* $\mathbf{X}_\mathcal{G}$. *Consider the output* $\mathcal{G}'$ *from applying findCausalClusters over* $\mathbf{X}_\mathcal{G}$. *If none of the clusters in* $\mathcal{G}'$ *is the bond set in* $\mathcal{G}$, *then all latent atomic covers have been correctly identified.*

However, without access to the true graph $\mathcal{G}$, we are not able to identify which clusters formed from *findCausalClusters* are bond sets. Fortunately, the following properties of the clusters formed over bond sets provide a way to remove bond sets in our discovered graph even without such knowledge, so that the clusters and latent covers can be correctly identified.

**Theorem 4** (Correcting Clusters). *Denote by $\mathcal{G}'$ the output from findCausalClusters and by $\mathcal{G}$ the true graph. For a latent atomic cover $\mathbf{L}'$ in $\mathcal{G}'$, if the measured pure descendants of $\mathbf{L}'$ is a bond set in the true graph $\mathcal{G}$, then there exist a set of siblings $\mathbf{S}$ of $\mathbf{L}'$ in $\mathcal{G}'$, a set of children $\mathbf{C}$ of $\mathbf{L}'$, and a set of grandparents $\mathbf{P}$ of $\mathbf{L}'$, such that $\mathcal{M}_{\mathcal{G}'}(\mathbf{S} \cup \mathbf{C} \cup \mathbf{P})$ forms a cluster that is not a bond set in $\mathcal{G}$.*

The above theorem shows that whenever forming a bond set, the siblings and grandparents of the latent cover are the key to correcting the incorrect clusters and covers. Specifically, we will remove such a cover and use its children to form new clusters with the siblings and grandparents (see details in Algorithm 3), and Theorem 4 guarantees that these new clusters will not contain bond sets, and thus, providing correct clusters.

**Example 3.** *In Figure 3(b), the formed cluster $\{X_9, X_{10}, X_{11}\}$ is the bond set in the true graph in Figure 3(a), and the covers $\{L_1', L_2'\}$, $\{L_7'\}$, and $\{L_8'\}$ are not correct. Illustration of the algorithm for this example, including refining the incorrect clusters and covers, is given in Appendix A5.7.*

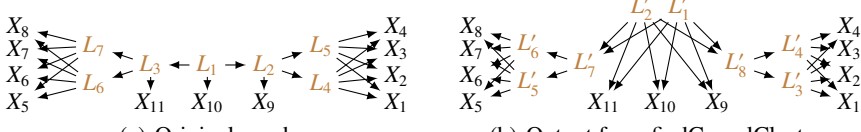

(a) Original graph          (b) Output from findCausalClusters

Figure 3: An example where findCausalClusters fails.

Furthermore, with Theorem 4, we give the following rule for correcting the clusters and covers.

> **Rule 2:** For each discovered latent atomic cover $\mathbf{L}$, let $\mathbf{V} = Gp_{\mathcal{G}'}(\mathbf{L}) \bigcup Sib_{\mathcal{G}'}(\mathbf{L}) \bigcup Ch_{\mathcal{G}'}(\mathbf{L})$ and apply *findCausalClusters* to $\mathbf{V}$ to refine the clusters.

Before introducing the detailed refining algorithm *refineClusters* based on Rule 2, we first introduce an operator *makeRoot* that will be used in the algorithm, which will not change the rank deficiency constraints (Lemma 5).

**Definition 8** (*makeRoot*). *Give a graph $\mathcal{G}'$ and a latent atomic cover $\mathbf{L}$ in $\mathcal{G}'$. A makeRoot operator of $\mathbf{L}$, denoted by makeRoot($\mathbf{L}$), reorients all outgoing edges of $\mathbf{L}$ to $\mathbf{L}$, such that $\mathbf{L}$ is a root variable.*

**Lemma 5** (Rank Invariance). *Denote by $\mathcal{G}'$ the output from findCausalClusters and by $\mathbf{L}$ a latent atomic cover in $\mathcal{G}'$. Then all rank constraints, that are possible to be executed by findCausalClusters prescribed by $\mathcal{G}'$, before and after the operator makeRoot($\mathbf{L}$) are identical.*

Algorithm 3 introduces the procedure of refining clusters based on Rule 2. Starting with the output $\mathcal{G}'$ from *findCausalClusters*, we proceed in a breadth-first search from the root variable (lines 1-3, 5). Because the root is trivially not over a bond set, Theorem 4 guarantees that for each child of the root, we will form new clusters that are not over bond sets. For each cover $\mathbf{L}$, if it has only one child, we will recursively consider the children of its child instead, since if $\mathbf{L}$ is over a bond set then its single child will also be. We add the first set of children into the search set $\mathbf{V}$ (line 4). We construct the search set (line 6) and remove the covers we are refining (lines 7-8). Finally, we make $\mathbf{L}$ the root (line 8) and conduct the search to form new clusters (line 9). The algorithm ends after refining every latent cover in $\mathcal{G}'$. With this refining procedure, we will derive the correct latent covers for the graph in Figure 3(a).

### 3.3 Phase III: Refining Edges

With *findCausalClusters* and *refineClusters*, the output $\mathcal{G}'$ correctly identifies the latent variables. Moreover, $\mathcal{G}'$ correctly identifies the following d-separation: for every $\mathbf{L} \in \mathbf{L}_{\mathcal{G}'}$, its parents $Pa_{\mathcal{G}'}(\mathbf{L})$ d-separates $\mathbf{L}$ and its descendants from the ancestors of $Pa_{\mathcal{G}'}(\mathbf{L})$, and thus, there cannot be any edges from each $\mathbf{L}$ to any of its ancestors beyond its own parents. However, $\mathcal{G}'$ may still have incorrect edges *locally*. Specifically, each time we create an atomic cover $\mathbf{L}$ in *findCausalClusters*, the implicit assumption is that each of its children is conditionally independent of the other children given the parents $\mathbf{L}$. However, this assumption is not necessarily true, as (i) the children of $\mathbf{L}$ may be directly connected to one another, and (ii) $\mathbf{L}$ may only be directly connected to a subset of its children. In other words, previous steps did not consider condition independence relationships across clusters. Hence, we need a further step to correct edges over each $\mathbf{L}$ and its children.

For example, consider the true graph in Figure 4(a), which ends up with the graph in Figure 4(b) with the first two phases. However, note that the first two phases did not consider the d-separation between $L_2'$ and $L_3'$, and thus the edges among $L_1', L_2', L_3', L_4'$ may not be correct, including the v structure. In

---

**Algorithm 3:** Phase II: refineClusters

---

**Input** : Output graph $\mathcal{G}'$ from Phase I
**Output** : Refined graph $\mathcal{G}'$

refineClusters $(\mathcal{G}')$:

1  Let $Q$ be an empty queue; $Q$.enqueue($PCh_{\mathcal{G}'}(Root(\mathcal{G}'))$);
2  **repeat**
3     $\mathbf{L} \leftarrow Q$.dequeue();
4     **while** $Ch_{\mathcal{G}'}(\mathbf{L})$ *is a single atomic cover* **do** remove $Ch_{\mathcal{G}'}(\mathbf{L})$ and add $Ch_{\mathcal{G}'}(Ch_{\mathcal{G}'}(\mathbf{L}))$ as children of $\mathbf{L}$ ;
5     **for** *each atomic cover* $C \in Ch_{\mathcal{G}'}(\mathbf{L})$ **do** $Q$.enqueue($C$) ;
6     $\mathbf{V} \leftarrow Gp_{\mathcal{G}'}(\mathbf{L}) \bigcup Sib_{\mathcal{G}'}(\mathbf{L}) \bigcup Ch_{\mathcal{G}'}(\mathbf{L})$ ;     // variables for finding new clusters
7     $\mathbf{R} \leftarrow \mathbf{L} \bigcup Pa_{\mathcal{G}}(\mathbf{L})$ ;            // atomic covers to be removed
8     $\mathcal{G}' \leftarrow$ makeRoot$_{\mathcal{G}'}(\mathbf{L})$; remove $\mathbf{R}$ and all adjoining edges from $\mathcal{G}'$;
9     $\mathcal{G}'' \leftarrow$ findCausalClusters($\mathbf{V}$); update $\mathcal{G}'$ with new clusters from $\mathcal{G}''$ ;  // find new clusters
10 **until** *Q is empty*;
11 **return** $\mathcal{G}'$

---

particular, in the true graph, $L_1$ d-separates $L_2$ from $L_3$, while $\{L_1, L_4\}$ does not d-separate $L_2$ from $L_3$, but these d-separations are not reflected in the discovered $\mathcal{G}'$ in the first two phases. Thus, we need to refine the edges. To this end, we first set $L'_1, L'_2, L'_3$ and $L'_2, L'_3, L'_4$ to be fully connected, and consider testing $\mathcal{A} = \{L'_2, X_1\}$ against $\mathcal{B} = \{L'_3, X_2\}$, where we partition the children of $L'_1$ into two sets and put them in $\mathcal{A}$ and $\mathcal{B}$, respectively. By doing so, we force $L'_1$ in $\mathcal{G}'$ to be in the separating set. Since $\text{rank}(\Sigma_{\mathcal{A},\mathcal{B}}) = 1$, it implies that no other variable is in the separating set, and therefore we can conclude that $L'_1$ d-separates $L'_2$ from $L'_3$. This principle is characterized in the following lemma.

**Lemma 6** (Cross-Cover Test). *Given a set of variables $\mathcal{S}$, consider two latent atomic covers $\mathbf{L}_A, \mathbf{L}_B \in \mathcal{S}$, and a potential separating set $\mathbf{C} = \{\mathbf{L}_{C_i}\} \subseteq \mathcal{S} \backslash \{\mathbf{L}_A, \mathbf{L}_B\}$. For each $\mathbf{L}_{C_i}$, consider $\mathbf{C}_i^A, \mathbf{C}_i^B \subseteq PCh(\mathbf{L}_{C_i})$ with $\mathbf{C}_i^A, \mathbf{C}_i^B \neq \emptyset$ and $\mathbf{C}_i^A \cap \mathbf{C}_i^B = \emptyset$, and denote the cardinality $k_i^A := min(|\mathbf{L}_{C_i}|, |\mathbf{C}_i^A|)$, $k_i^B := min(|\mathbf{L}_{C_i}|, |\mathbf{C}_i^B|)$, respectively. Then there is no edge between $\mathbf{L}_A$ and $\mathbf{L}_B$ if and only if there exists a separating set $\mathbf{C}$ such that $\text{rank}(\Sigma_{\mathcal{A},\mathcal{B}}) < min(|\mathbf{L}_A| + \sum_i k_i^A, |\mathbf{L}_B| + \sum_i k_i^B)$, where $\mathcal{A} = \{\mathbf{L}_A, \mathbf{C}_1^A, \mathbf{C}_2^A, ...\}$ and $\mathcal{B} = \{\mathbf{L}_B, \mathbf{C}_1^B, \mathbf{C}_2^B, ...\}$. In this case, we say that $\mathbf{C}$ satisfies the cross-cover test of $\mathbf{L}_A$ against $\mathbf{L}_B$.*

Note that in order to find rank deficiency when performing the cross-cover test, $\mathbf{L}_{C_i}$ needs to satisfy the third condition in Condition 1. Based on Lemma 6, we use the following rule to refine the edges.

    **Rule 3:** For a pair of latent covers $(\mathbf{L}_A, \mathbf{L}_B)$, let $\mathcal{A} \leftarrow \{\mathbf{L}_A, \mathbf{C}_1^A, \mathbf{C}_2^A, ...\}$ and $\mathcal{B} \leftarrow \{\mathbf{L}_B, \mathbf{C}_1^B, \mathbf{C}_2^B, ...\}$. If there exists such $\mathcal{A}, \mathcal{B}$ such that $\text{rank}_{\mathcal{G}'}(\Sigma_{\mathcal{A},\mathcal{B}})$ is rank deficient, then remove all edges between $\mathbf{L}_A, \mathbf{L}_B$ in $\mathcal{G}'$.

Furthermore, with Rule 3, Algorithm 4a (*CrossCoverTest*) in Appendix A gives the procedure of refining the edges over a set of latent variables $\mathcal{S}$ to correct the causal skeleton.

Furthermore, we are going to identify the v structures among latent atomic covers. In the output $\mathcal{G}'$ from phase II, for any $\mathbf{L}'$ with a child $\mathbf{C}'_i$ and parent $\mathbf{P}'$, it is not possible to have a collider $\mathbf{C}'_i \rightarrow \mathbf{L}' \leftarrow \mathbf{P}'$ in the ground-truth graph, because in this case the cluster would not have been rank deficient. Therefore, the only v structures in the graph are amongst the variables in $\mathbf{L}' \cup PCh_{\mathcal{G}'}(\mathbf{L}')$, and similar to *crossCoverTest*, we only need to test for v structures locally. Continue to consider the example in Figure 4. The edge between $L'_2$ and $L'_3$ is missing because the rank over $\{L'_2, X_1\}$ and $\{L'_3, X_2\}$ is 1, implying that $L'_1$ d-separates $L'_2$ from $L'_3$, as is done in *crossCoverTest*. Now, since $L'_2 - L'_4 - L'_3$ forms an unshielded triplet, we want to test if a collider exists at $L'_4$. We find that the rank over $\mathcal{A} = \{L'_2, L'_4, X_1\}$ and $\mathcal{B} = \{L'_3, X_2\}$ is $2 > 1$, and the rank over $\mathcal{A} = \{L'_2, X_1\}$ and $\mathcal{B} = \{L'_3, L'_4, X_2\}$ is $2 > 1$, so $L'_2 \rightarrow L'_4 \leftarrow L'_3$. For general cases, the rule for finding v structures are formulated in the following lemma and Algorithm 4b (*findColliders*) in Appendix A.

**Lemma 7** (V-Structure Test). *For any unshielded triangle $\mathbf{L}_A - \mathbf{L}_C - \mathbf{L}_B$, let $\mathcal{A}, \mathcal{B}$ be the set of variables in Lemma 6 such that $\Sigma_{\mathcal{A},\mathcal{B}}$ was rank deficient. Let $k = \text{rank}(\Sigma_{\mathcal{A},\mathcal{B}})$, $k_1 = \text{rank}(\Sigma_{\mathcal{A} \cup \mathbf{L}_C,\mathcal{B}})$, and $k_2 = \text{rank}(\Sigma_{\mathcal{A},\mathcal{B} \cup \mathbf{L}_C})$. Then, $\mathbf{L}_A \rightarrow \mathbf{L}_B \leftarrow \mathbf{L}_C$ if and only if $k < min(k_1, k_2)$.*

As mentioned above, we only need to perform cross-cover test and v-structure test locally in the estimated graph $\mathcal{G}'$. Algorithm 4 (Phase III: *refineEdges*) combines the search procedure of the two components together with the output $\mathcal{G}'$ from *refineClusters* as the input. Specifically, for each latent cover $\mathbf{L}'$ in $\mathcal{G}'$, we only need to consider testing for edges amongst $\mathbf{L}'$ and its children $\mathbf{C} = Ch_{\mathcal{G}'}(\mathbf{L}')$. Thus, we perform a depth-first traversal of the output graph $\mathcal{G}'$ starting from $Root(\mathcal{G}')$ (lines 1-3), and

apply the cross-cover test at each **L** to refine the skeleton of $\mathcal{G}'$ (lines 4-7, 11) and the collider test to identify the v structures (lines 10, 11). After determining v-structures, we can find more directions by applying Meek's rule (line 12), analogous to that in the PC algorithm [Spirtes et al., 2000].

---

**Algorithm 4:** Phase III: refineEdges

**Input** : Learned graph $\mathcal{G}'$ from phase II
**Output** : Markov equivalence class $\mathcal{G}'$
refineEdges ($\mathcal{G}'$):
1 **foreach** *latent atomic cover* **L**′ *in* $\mathcal{G}'$ **do**
2     **if** **L**′ *does not have latent children* **then return** $\mathcal{G}'$ ;
3     **foreach** *latent child* $\mathbf{C}_i$ *of* **L**′ **do** $\mathcal{G}' \leftarrow$ refineEdges($\mathcal{G}', \mathbf{C}_i$) ;
4     **if** $\mathbf{C} := PCh_{\mathcal{G}'}(L')$ *is a single latent cover* **then** $\mathcal{S} \leftarrow \mathbf{L}' \cup \mathbf{C} \cup PCh_{\mathcal{G}'}(\mathbf{C})$ ;
5     **else** $\mathcal{S} \leftarrow \mathbf{L}' \cup \mathbf{C}$ ;
6     $\mathcal{G}'' \leftarrow$ makeRoot$_{\mathcal{G}'}$(**L**′) and remove all edges amongst $\mathcal{S}$ in $\mathcal{G}''$ ;           // temp graph
7     Edgeset $\mathcal{E} \leftarrow$ crossCoverTest($\mathcal{S}, \mathcal{G}''$);
8     **if** *no conditional independencies found* **then return** $\mathcal{G}'$;
9     **else**
10         Collider set $C \leftarrow$ findColliders($\mathcal{S}, \mathcal{E}, \mathcal{G}''$);
11         in $\mathcal{G}'$, remove all edges amongst $\mathcal{S}$ and use $\mathcal{E}, C$ to connect variables in $\mathcal{S}$;
12         apply Meek's rule to $\mathcal{G}'$;
13 convert $\mathcal{G}'$ to its Markov equivalence class;
14 **return** $\mathcal{G}'$

---

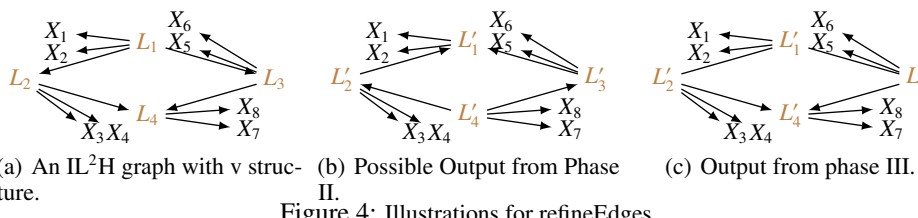

(a) An IL$^2$H graph with v structure.     (b) Possible Output from Phase II.     (c) Output from phase III.

Figure 4: Illustrations for refineEdges.

# 4 Theoretical Results

In this section, we show the correctness of the algorithms proposed in Section 3. In particular, by making use of the rank constraints of only the measured variables, the proposed algorithms output the correct Markov equivalence class of the IL$^2$H graph asymptotically, under the *minimal-graph operator* and *skeleton operator*, with their definitions given below.

**Definition 9** (Markov Equivalence Class of IL$^2$H graphs). *Two IL$^2$H graphs $\mathcal{G}_1$ and $\mathcal{G}_2$ are in the same Markov equivalence class, denoted by $\mathcal{G}_1 \approx \mathcal{G}_2$, if and only (1) they have the same set of variables (both measured and latent variables), (2) have the same causal skeleton, and (3) have the same V-structures $\mathbf{L}_i \rightarrow \mathbf{L}_k \leftarrow \mathbf{L}_j$, where $\mathbf{L}_i, \mathbf{L}_j, \mathbf{L}_k$ represent latent atomic covers.*

**Definition 10** (Minimal-Graph Operator). *Suppose $\mathcal{G}$ is an IL$^2$H graph. For every latent atomic cover $\mathbf{L}$ in $\mathcal{G}$, merge $\mathbf{L}$ to its parents $\mathbf{P}$ if the following conditions hold: (1) $\mathbf{L}$ is the pure children of $\mathbf{P}$, (2) $|\mathbf{L}| = |\mathbf{P}|$, and (3) the pure children of $\mathbf{L}$ form one latent atomic cover, or the siblings of $\mathbf{L}$ form one latent atomic cover. We call such operator the minimal-graph operator and denote it by $O_{min}(\mathcal{G})$.*

**Definition 11** (Skeleton Operator). *Suppose $\mathcal{G}$ is an IL$^2$H graph. A skeleton operator of $\mathcal{G}$, denoted by $O_s(\mathcal{G})$ is defined as follows: for any latent atomic cover $\mathbf{L}$, draw an edge from $l_j \in \mathbf{L}$ to $c_k \in PCh_{\mathcal{G}}(\mathbf{L}_i)$, if $l_j$ and $c_k$ are not directly connected in $\mathcal{G}$.*

Note that the minimal-graph operator and the skeleton operator will not change the rank constraints, or in other words, graphs before and after applying the operators are indistinguishable with rank constraints. This result is shown in the following lemma.

**Lemma 8.** *Suppose $\mathcal{G}$ is an IL$^2$H graph. The rank constraints are invariant with the minimal-graph operator and the skeleton operator; that is, $\mathcal{G}$ and $O_{skeleton}(O_{min}(\mathcal{G}))$ are rank equivalent.*

These two operators have already been achieved in Algorithm 1, so the output of the algorithm is the rank-equivalent graph with the minimal number of latent atomic covers.

We next proceed to show that phases I-III will output a graph $\mathcal{G}'$ such that $\mathcal{G}'$ will be in the same Markov equivalence class as $O_{min}(O_s(\mathcal{G}))$, denoted by $\mathcal{G}' \approx \mathcal{G}$. We have already shown earlier in

Theorem 3 that *findCausalClusters* gives correct latent covers when there is no bond set. Furthermore, Theorem 4 shows that even in the presence of bond sets, Phase II *refineClusters* is able to refine clusters into those without bond sets. Therefore, Phase I-II can correctly identify the latent atomic covers of $O_{min}(\mathcal{G})$, which is given in the following theorem.

**Theorem 9** (Identifiability of Latent Variables). *Suppose $\mathcal{G}$ is an $IL^2H$ graph with measured variables $\mathbf{X}_\mathcal{G}$. Phases I-II in Algorithm 1 over $\mathbf{X}_\mathcal{G}$ can asymptotically identify the latent atomic covers of $O_{min}(\mathcal{G})$, with only the first two conditions in Condition 1.*

Moreover, Phase III *refineEdges* further guarantees correct skeletons and v structures. Therefore, the following theorem shows that Algorithm 1, which includes Phases I-III, can asymptotically identify the Markov equivalence class, up to the skeleton operator and the minimal-graph operator.

**Theorem 10** (Identifiability of Causal Graph). *Suppose $\mathcal{G}$ is an $IL^2H$ graph with measured variables $\mathbf{X}_\mathcal{G}$. Algorithm 1, including Phases I-III, over $\mathbf{X}_\mathcal{G}$ can asymptotically identify the Markov equivalence class of $O_{min}(O_s(\mathcal{G}))$.*

## 5 Experimental Results

We applied the proposed algorithm to synthetic data to learn the latent hierarchical causal graph. Specifically, we considered different types of latent graphs and different sample sizes. The causal strength was generated uniformly from $[-5, -0.5] \cup [0.5, 5]$, and the noise term was randomly chosen from a normal distribution with noise variance uniformly sampled from $[1, 5]$. To the best of our knowledge, this is the first algorithm that can identify such general latent hierarchical structures, so to fairly compare with other methods, besides general $IL^2H$ graphs, we also considered tree structures and measurement models. We compared the proposed method with tree-based method–Chow-Liu Recursive Grouping (CLRG) [Choi et al., 2011], as well as measurement-model-based methods, including FOFC [Kummerfeld and Ramsey, 2016] and GIN [Xie et al., 2020].

Table 1 gives the estimation results evaluated on three types of latent graphs: $IL^2H$ graphs, tree structures, and measurement models. The performance is measured by the percentage of correctly identified causal clusters over only measured variables. Our method gives the best results on all types of graphs, indicating that it can handle not only the tree-based and measurement-based structures, but also the latent hierarchical structure. The CLRG algorithm does not perform well on tree-based structure because the metric is rather strict–even a single misclustered variable outputs an error. Complete experimental settings and more results, including performance measured by other metrics and the case when the noise terms are uniformly distributed, are given in Appendix B.

Table 1: Performance (mean (standard deviation)) on learning different types of latent graphs.

| Sample size | $IL^2H$ | | | | Tree | | | | Measurement model | | | |
|---|---|---|---|---|---|---|---|---|---|---|---|---|
| | Ours | CLRG | FOFC | GIN | Ours | CLRG | FOFC | GIN | Ours | CLRG | FOFC | GIN |
| 2k | 0.70 (0.22) | 0 | 0.12 | 0.35 | 0.89 (0.12) | 0 | 0.38 | 0.13 | 0.92 (0.08) | 0 | 0.38 | 0.3 |
| 5k | 0.83 (0.15) | 0 | 0.17 | 0.40 | 1.0 (0.00) | 0 | 0.75 | 0.23 | 1.0 (0.00) | 0 | 0.65 | 0.7 |
| 10k | 0.86 (0.13) | 0 | 0.29 | 0.44 | 1.0 (0.00) | 0.13 | 0.87 | 0.50 | 1.0 (0.00) | 0 | 1.0 | 0.7 |

## 6 Conclusions and Future Work

In this paper, we formulated a specific type of latent hierarchical causal model and proposed a method to identify its graph by making use of rank deficiency constraints. Theoretically, we show that the proposed algorithm can find the correct Markov equivalence class of the whole graph asymptotically under mild restrictions of the graph structure. For more general graphs, only using the second-order statistics may result in a rank equivalence class that contains multiple DAGs, so how to further leverage high-order statistics to distinguish between causal graphs within the equivalence class will be our future work. Other future research directions include allowing nonlinear causal relationships and allowing measured variables to cause latent variables (existing techniques, e.g., Adams et al. [2021], Squires et al. [2022], may help to mitigate this issue).

## Acknowledgement

BH would like to acknowledge the support of Apple Scholarship. Kun Zhang was partially supported by the National Institutes of Health (NIH) under Contract R01HL159805, by the NSF-Convergence Accelerator Track-D award #2134901, by a grant from Apple Inc., and by a grant from KDDI Research Inc. FX acknowledges the support by National Natural Science Foundation of China (NSFC 62006051).

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
