# Appendix for "Latent Hierarchical Causal Structure Discovery with Rank Constraints"

**Biwei Huang** [*1] **Charles Low** [*1]**, Feng Xie**[3]**, Clark Glymour**[1]**, Kun Zhang**[1,2]

[1] Carnegie Mellon University
[2] Mohamed bin Zayed University of Artificial Intelligence
[3] Beijing Technology and Business University, China
{bwei.huang, charleslow88, xiefeng009}@gmail.com,
cg09@andrew.cmu.edu, kunz1@cmu.edu

Organization of Appendix:

- Section A: Algorithms that are not listed in the main text because of page limits
- Section B: Complete experimental results
- Section C: Proofs
- Section D: Related work
- Section E: More explanations on identifiability conditions, algorithms, and theorems
- Section F: Illustrative examples of the entire algorithm

| Pa: Parents | Sib: Siblings | $\mathbf{V}_{\mathcal{G}}$: All variables in graph $\mathcal{G}$ |
|---|---|---|
| PCh: Pure children | $\mathcal{M}$: Measured pure descendants | $\mathbf{X}$: A set of measured variables |
| PDe: Pure descendants | $\mathbf{X}_{\mathcal{G}}$: All measured variables in graph $\mathcal{G}$ | $\mathbf{L}$: A set of latent variables |
| Gp: Grandparents | $\mathbf{L}_{\mathcal{G}}$: All latent variables in graph $\mathcal{G}$ | $\mathbf{V}$: A set of variables |

Table 3: Complete graphical notations used in the paper.

## A    Algorithms on *crossCoverTest* and *findColliders*

Algorithm 4a: (*CrossCoverTest*) gives the procedure of refining the edges over a set of latent variables $\mathcal{S}$. It first fully connects the latent covers in $\mathcal{S}$ (line 2). Then for every pair of latent covers, it performs the cross-cover test (lines 3-17). If rank deficiency is found, then remove the corresponding edges (lines 10-12).

Algorithm 4b: (*findColliders*) gives the procedure of finding v structures. For every unshielded triangle $\mathbf{L}_1 - \mathbf{L}_3 - \mathbf{L}_2$, it performs v structure tests and compares with the rank which does not involve $\mathbf{L}_3$ (lines 22-23). If the rank that involves $\mathbf{L}_3$ is larger, then the unshielded triangle forms a v structure (lines 24-25).

## B    Complete Experimental Results

We applied the proposed algorithm to synthetic data to learn the latent hierarchical causal graph. Specifically, we considered different types of latent graphs and different sample sizes (with $N = 2k, 5k, 10k$). The causal strength was generated uniformly from $[-5, -0.5] \cup [0.5, 5]$, and the noise term either follows a Gaussian distribution (with noise variance uniformly sampled from $[1, 5]$) or a uniform distribution $\mathcal{U}(-2, 2)$.

---

[*]These authors contributed equally to this work.

36th Conference on Neural Information Processing Systems (NeurIPS 2022).

---
**Algorithm 4a:** crossCoverTest

**Input** : A set of latent variables $\mathcal{S}$, currently learned graph $\mathcal{G}'$
**Output** : An edgeset $\mathcal{E}$ among the variables

crossCoverTest $(\mathcal{S}, \mathcal{G}')$:

1   $\mathcal{E} \leftarrow \emptyset$;
2   add undirected edges between every pair of latent atomic covers in $\mathcal{S}$ to $\mathcal{E}$;
3   **foreach** *pair of latent atomic covers* $\mathbf{L}_A, \mathbf{L}_B \in \mathcal{S}$ **do**
4     k = 0
5     **repeat**
6       **repeat**
7         draw a potential separating set of $k$ atomic covers $\mathbf{C} = \{\mathbf{C}_1, \mathbf{C}_2, \cdots, \mathbf{C}_k\} \subseteq \mathcal{S} \backslash \{\mathbf{L}_A, \mathbf{L}_B\}$;
         **foreach** *atomic cover* $\mathbf{C}_i \in \mathbf{C}$ **do**
8           partition $PCh_{\mathcal{G}'}(\mathbf{C}_i)$ into $\mathbf{C}_i^A, \mathbf{C}_i^B$ ;            // Remark on clever choice
9         $\mathcal{A} \leftarrow \{\mathbf{L}_A, \mathbf{C}_1^A, \mathbf{C}_2^A, ...\}$ and $\mathcal{B} \leftarrow \{\mathbf{L}_B, \mathbf{C}_1^B, \mathbf{C}_2^B, ...\}$;
10        **if** *there exists such* $\mathcal{A}, \mathcal{B}$ *such that* $\mathrm{rank}_{\mathcal{G}'}(\Sigma_{\mathcal{A}, \mathcal{B}})$ *is rank deficient* **then**
11          remove all edges between $\mathbf{L}_A, \mathbf{L}_B$ in $\mathcal{E}$;
12          break;
13       **until** *all sets* $\mathbf{C}$ *with $k$ atomic covers tested*;
14       **if** *rank deficiency found* **then**
15         break;
16       $k \mathrel{+}= 1$;
17     **until** $k >$ *number of variables in* $\mathcal{S} \backslash \{\mathbf{L}_A, \mathbf{L}_B\}$;
18   **return** *Edgeset* $\mathcal{E}$

---

---
**Algorithm 4b:** findColliders

**Input** : A set of latent variables $\mathcal{S}$, edgeset $\mathcal{E}$, currently learned graph $\mathcal{G}'$
**Output** : A set of colliders $C$

findColliders $(\mathcal{S}, \mathcal{E}, \mathcal{G}')$:

19   Collider set $C \leftarrow \emptyset$;
20   **foreach** *unshielded triangle* $\mathbf{L}_1 - \mathbf{L}_3 - \mathbf{L}_2$ *in* $\mathcal{S}$ *based on* $\mathcal{E}$ **do**
21     let $\mathcal{A}, \mathcal{B}$ be the set of variables in Algorithm 4a: such that $\Sigma_{\mathcal{A}, \mathcal{B}}$ was rank deficient with rank $k$;
22     $k_1 \leftarrow \mathrm{rank}_{\mathcal{G}'}(\Sigma_{\mathcal{A} \cup \mathbf{L}_3, \mathcal{B}})$;
23     $k_2 \leftarrow \mathrm{rank}_{\mathcal{G}'}(\Sigma_{\mathcal{A}, \mathcal{B} \cup \mathbf{L}_3})$;
24     **if** $k < \min(k_1, k_2)$ **then**
25       add collider $\mathbf{L}_1 \rightarrow \mathbf{L}_3 \leftarrow \mathbf{L}_2$ to $C$;
26   **return** *Collider set* $C$

---

To the best of our knowledge, this is the first algorithm that can identify such general latent hierarchical structures, so to fairly compare with other methods, besides general IL²H graphs (see Figure 7), we also considered tree structures (see Figure 5) and measurement models (see Figure 6). We compared the proposed method with the tree-based method–Chow-Liu Recursive Grouping (CLRG) [Choi et al., 2011], as well as measurement-model-based methods, including FOFC [Kummerfeld and Ramsey, 2016] and GIN Xie et al. [2020].

We used the following metrics to evaluate the performance:

- *Causal cluster recovery rate over measured variables (metric 1)*: measured by the percentage of correctly identified causal clusters over measured variables, with

$$m_1 = \frac{\text{correctly found \# clusters over measured variables}}{\text{total \# clusters over measured variables}}.$$

- *Causal cluster recovery rate over all variables (metric 2)*, measured by the percentage of correctly identified causal clusters over all variables, with

$$m_2 = \frac{\text{correctly found \# clusters over all variables}}{\text{total \# clusters over all variables}}.$$

- *Percentage differences between estimated and true adjacency matrices (metric 3)*, with

$$m_3 = \sum_{i,j} (Adj_{\mathcal{G}}(i, j) \sim= Adj_{\mathcal{G}'}(i, j))/((n_{\mathbf{X}} + n_{\mathbf{L}})^2 - n_{\mathbf{X}}^2),$$

where *Adj* denotes the adjacency matrix, $i$ and $j$ denote the $i$-th and $j$-th entry, respectively, and $n_{\mathbf{X}}$ and $n_{\mathbf{L}}$ are the number of measured variables and latent variables, respectively. Note that the indices of the latent variables in the estimated graph may not be aligned to those in the true graph. To remove this ambiguity, we tried all permutations of the latent indices in the estimated graph and used the one which has the smallest difference from the true graph. Moreover, if the estimated number of latent variables is smaller than the true number of latent variables, add extra latent variables to $\mathcal{G}'$ that do not have edges with others. If the estimated number of latent variables is larger than the true number of latent variables, then find a subset of the latent variables in $\mathcal{G}'$ that best aligns the true ones.

It is worth mentioning that how to measure the performance of the estimated latent hierarchical graph is a nontrivial problem and will be further investigated.

The experimental results were reported in Tables 4 and 5, where the noise terms are Gaussian distributed and uniformly distributed, respectively. Our method gives the best results on all types of graphs, indicating that it can handle not only the tree-based and measurement-based structures, but also the latent hierarchical structure. The CLRG algorithm does not perform well on tree-based structure because the first two metrics are rather strict–even a single mis-clustered variable outputs an error.

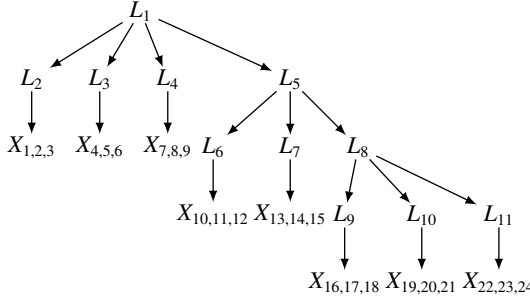

Figure 5: Tree. Note that $X_{i,j,k}$ means $X_i, X_j, X_k$.

Figure 6: Measurement model.

## C  Related Work

Identification of causal relationships from observational data, known as *causal discovery*, is attractive for the reason that traditional randomized control trials may be hard or even impossible to do. Most state-of-the-art approaches in causal discovery assume that the measured variables are the underlying causal variables and that no latent confounders influence the measured variables [Spirtes et al., 2000, Chickering, 2002, Shimizu et al., 2006, Hoyer et al., 2009, Zhang and Hyvärinen, 2009]. However, in many real-world problems, this assumption may not hold.

For example, in complex systems, it is usually hard to enumerate and measure all task-related variables, so there may exist latent variables that influence multiple observed variables, the ignorance of which may introduce spurious correlations between measured variables. A more complex scenario is that the variables form a hierarchical structure, where the latent variables may generate latent variables in a hierarchical way, while only the leaf nodes are measured, which is common in real-world scenarios. For instance, in fMRI data analysis, hundreds of thousands of voxels are recorded, where these micro-variables may not be necessary to have clear semantic meaning. Therefore, from the measured voxels, we aim to automatically identify conceptually meaningful functional brain regions of different levels, where the lower level represents simpler functional regions and the higher level represents more abstract and complex functional regions, which thus form a hierarchical structure.

Table 4: Performance (mean (standard deviation)) on learning different types of latent graphs, where noise terms were generated from Gaussian distributions.

| Algorithm | | metric 1 ↑ | | | |
|---|---|---|---|---|---|
| | | Ours | CLRG | FOFC | GIN |
| $IL^2H$ | 2k | 0.70 (0.22) | 0.00 (0.00) | 0.12 (0.09) | 0.35 (0.23) |
| | 5k | 0.83 (0.15) | 0.00 (0.00) | 0.17 (0.10) | 0.40 (0.24) |
| | 10k | 0.86 (0.13) | 0.00 (0.00) | 0.29 (0.10) | 0.44 (0.21) |
| Tree | 2k | 0.89 (0.12) | 0.00 (0.00) | 0.38 (0.25) | 0.13 (0.30) |
| | 5k | 1.0 (0.00) | 0.00 (0.00) | 0.75 (0.23) | 0.23 (0.30) |
| | 10k | 1.0 (0.00) | 0.13 (0.04) | 0.87 (0.20) | 0.50 (0.00) |
| Measurement Model | 2k | 0.92 (0.08) | 0.00 (0.00) | 0.38 (0.22) | 0.30 (0.16) |
| | 5k | 1.0 (0.00) | 0.00 (0.00) | 0.65 (0.32) | 0.70 (0.43) |
| | 10k | 1.0 (0.00) | 0.00 (0.00) | 1.0 (0.00) | 0.70 (0.43) |

| Algorithm | | metric 2 ↑ | | | |
|---|---|---|---|---|---|
| | | Ours | CLRG | FOFC | GIN |
| $IL^2H$ | 2k | 0.60 (0.16) | 0.00 (0.00) | 0.09 (0.07) | 0.26 (0.19) |
| | 5k | 0.69 (0.22) | 0.00 (0.00) | 0.09 (0.07) | 0.28 (0.19) |
| | 10k | 0.73 (0.17) | 0.00 (0.00) | 0.12 (0.07) | 0.35 (0.17) |
| Tree | 2k | 0.79 (0.19) | 0.00 (0.00) | 0.28 (0.19) | 0.09 (0.11) |
| | 5k | 0.83 (0.16) | 0.00 (0.00) | 0.55 (0.17) | 0.17 (0.22) |
| | 10k | 0.89 (0.09) | 0.09 (0.03) | 0.63 (0.18) | 0.36 (0.20) |
| Measurement Model | 2k | 0.92 (0.08) | 0.00 (0.00) | 0.38 (0.22) | 0.30 (0.16) |
| | 5k | 1.0 (0.00) | 0.00 (0.00) | 0.65 (0.32) | 0.70 (0.43) |
| | 10k | 1.0 (0.00) | 0.00 (0.00) | 1.0 (0.00) | 0.70 (0.43) |

| Algorithm | | metric 3 ↓ | | | |
|---|---|---|---|---|---|
| | | Ours | CLRG | FOFC | GIN |
| $IL^2H$ | 2k | 0.11 (0.02) | 0.18 (0.04) | 0.26 (0.21) | 0.16 (0.03) |
| | 5k | 0.10 (0.02) | 0.18 (0.04) | 0.26 (0.21) | 0.15 (0.03) |
| | 10k | 0.10 (0.02) | 0.18 (0.04) | 0.20 (0.27) | 0.15 (0.03) |
| Tree | 2k | 0.02 (0.00) | 0.10 (0.00) | 0.09 (0.00) | 0.15 (0.02) |
| | 5k | 0.02 (0.00) | 0.10 (0.00) | 0.09 (0.00) | 0.10 (0.02) |
| | 10k | 0.01 (0.00) | 0.09 (0.00) | 0.08 (0.00) | 0.09 (0.00) |
| Measurement Model | 2k | 0.02 (0.00) | 0.28 (0.00) | 0.18 (0.04) | 0.30 (0.08) |
| | 5k | 0.00 (0.00) | 0.28 (0.00) | 0.13 (0.05) | 0.17 (0.15) |
| | 10k | 0.00 (0.00) | 0.28 (0.00) | 0.00 (0.00) | 0.17 (0.15) |

Note: ↑ means a higher value is better, and vice versa.

We may also see similar structures in image representation learning–image pixels are dependent, and it seems sensible to consider them as observations generated by multiple-layer hidden concepts.

Previous causal discovery approaches that can handle latent confounders are mainly based on the following criteria.

- Conditional independence constraints. The FCI algorithm [Spirtes et al., 2000], as well as its variants [Colombo et al., 2012, Pearl, 2000, Akbari et al., 2021], makes use of conditional independence tests over observed variables to identify the causal structure over observed variables up to a maximal ancestral graph. This type of methods can handle both linear and nonlinear causal relationships, but the limitation is that there are large indeterminacies in the resulting graph about the existence of an edge, as well as the existence of confounders. In practice, it is often the case that the resulting graph contains many undetermined edges, denoted by ∘–∘, where the circle can be either tail or arrow. Moreover, they do not consider the causal relationships among latent variables.

- Tetrad condition. With the Tetrad condition, i.e., the rank constraints of every $2 \times 2$ off-diagonal sub-covariance matrix, one is able to locate latent variables and identify the causal skeleton among them in linear-Gaussian models [Silva et al., 2006, Kummerfeld and Ramsey, 2016, Wang, 2020]. These methods assume that each observed variable is influenced by only one latent parent, and

Table 5: Performance (mean (standard deviation)) on learning different types of latent graphs, where noise terms were generated from uniform distributions.

| | | metric 1 ↑ | | | |
|---|---|---|---|---|---|
| Algorithm | | Ours | CLRG | FOFC | GIN |
| $IL^2H$ | 2k | 0.72 (0.19) | 0.00 (0.00) | 0.10 (0.10) | 0.45 (0.18) |
| | 5k | 0.87 (0.13) | 0.00 (0.00) | 0.17 (0.10) | 0.46 (0.18) |
| | 10k | 0.88 (0.10) | 0.00 (0.00) | 0.29 (0.10) | 0.52 (0.16) |
| Tree | 2k | 0.93 (0.07) | 0.00 (0.00) | 0.40 (0.27) | 0.79 (0.22) |
| | 5k | 1.0 (0.00) | 0.00 (0.00) | 0.75 (0.23) | 0.99 (0.04) |
| | 10k | 1.0 (0.00) | 0.00 (0.00) | 0.87 (0.20) | 0.95 (0.08) |
| Measurement Model | 2k | 0.95 (0.06) | 0.00 (0.00) | 0.25 (0.21) | 0.87 (0.20) |
| | 5k | 1.0 (0.00) | 0.00 (0.00) | 0.70 (0.27) | 1.00 (0.00) |
| | 10k | 1.0 (0.00) | 0.00 (0.00) | 1.0 (0.00) | 1.00 (0.00) |

| | | metric 2 ↑ | | | |
|---|---|---|---|---|---|
| Algorithm | | Ours | CLRG | FOFC | GIN |
| $IL^2H$ | 2k | 0.61 (0.20) | 0.00 (0.00) | 0.07 (0.07) | 0.34 (0.15) |
| | 5k | 0.71 (0.19) | 0.00 (0.00) | 0.09 (0.07) | 0.35 (0.15) |
| | 10k | 0.76 (0.20) | 0.00 (0.00) | 0.12 (0.07) | 0.39 (0.14) |
| Tree | 2k | 0.83 (0.12) | 0.00 (0.00) | 0.29 (0.20) | 0.57 (0.16) |
| | 5k | 0.85 (0.11) | 0.00 (0.00) | 0.55 (0.17) | 0.72 (0.03) |
| | 10k | 0.89 (0.15) | 0.00 (0.00) | 0.63 (0.18) | 0.69 (0.06) |
| Measurement Model | 2k | 0.95 (0.06) | 0.00 (0.00) | 0.25 (0.21) | 0.87 (0.20) |
| | 5k | 1.0 (0.00) | 0.00 (0.00) | 0.70 (0.27) | 1.00 (0.00) |
| | 10k | 1.0 (0.00) | 0.00 (0.00) | 1.0 (0.00) | 1.00 (0.00) |

| | | metric 3 ↓ | | | |
|---|---|---|---|---|---|
| Algorithm | | Ours | CLRG | FOFC | GIN |
| $IL^2H$ | 2k | 0.10 (0.02) | 0.18 (0.04) | 0.23 (0.21) | 0.15 (0.03) |
| | 5k | 0.10 (0.02) | 0.18 (0.04) | 0.26 (0.21) | 0.15 (0.03) |
| | 10k | 0.10 (0.02) | 0.18 (0.04) | 0.20 (0.27) | 0.14 (0.03) |
| Tree | 2k | 0.02 (0.00) | 0.11 (0.00) | 0.10 (0.00) | 0.11 (0.00) |
| | 5k | 0.01 (0.00) | 0.10 (0.00) | 0.09 (0.00) | 0.11 (0.00) |
| | 10k | 0.01 (0.00) | 0.10 (0.00) | 0.08 (0.00) | 0.11 (0.00) |
| Measurement Model | 2k | 0.01 (0.00) | 0.28 (0.00) | 0.25 (0.04) | 0.07 (0.05) |
| | 5k | 0.00 (0.00) | 0.28 (0.00) | 0.25 (0.04) | 0.07 (0.05) |
| | 10k | 0.00 (0.00) | 0.28 (0.00) | 0.00 (0.00) | 0.00 (0.00) |

Note: ↑ means a higher value is better, and vice versa.

each latent variable has at least three pure measured children. Moreover, the Tetrad condition can also be used to identify a latent tree structure [Pearl, 1988].

- Matrix decomposition. It has been shown that, under certain conditions, the precision matrix can be decomposed into a low-rank matrix and a sparse matrix, where the low-rank matrix represents the causal structure from latent variables to observed variables and the sparse matrix gives the structural relationships over observed variable. To achieve such decomposition, however, certain assumptions are imposed on the structure [Chandrasekaran et al., 2011, 2012]. A related work [Anandkumar et al., 2013] decomposed the covariance matrix into a low-rank matrix and a diagonal matrix, which requires three times more measured variables than latent variables. [Anandkumar et al., 2013] can also handle multi-level DAGs that some latent variables do not have measured variables as children, but it requires that the underlying graph can be partitioned into multiple levels such that all the edges are between nodes in adjacent layers; the graphs in Figure 1(b) in the main text, and Figures 5-6 and Figures 7(a, b, d) in Appendix are not satisfied.

- Over-complete independent component analysis (ICA)-based methods. Several methods [Shimizu et al., 2009] make use of over-complete ICA to learn the causal structure with latent variables, since it allows more source signals than observed variables. These methods do not consider the causal structure among latent variables and the size of the equivalence class of the identified structure could be large. In addition, in practice, the estimation of over-complete ICA models is easy to

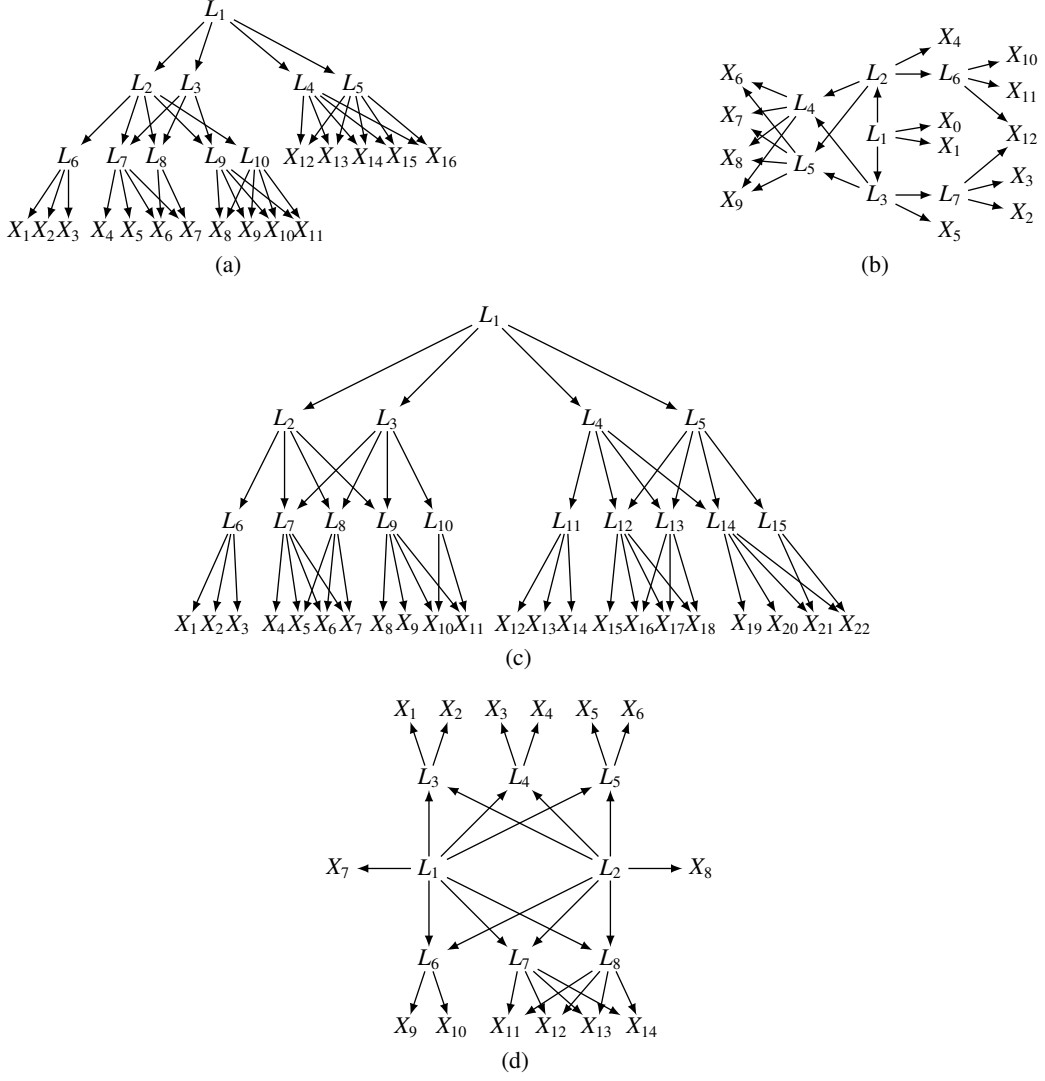

Figure 7: Example IL$^2$H graphs.

get stuck in local optima, unless the underlying sources are very sparse [Entner and Hoyer, 2010, Tashiro et al., 2014].

- Generalized independent noise (GIN) condition. The GIN condition is an extension of the independent noise condition in the existence of latent confounders. It assumes the noise terms are non-Gaussian and leverages higher-order statistics to identify latent structures. In particular, Xie et al. [2020] proposes a GIN-based approach that allows multiple latent parents behind every pair of observed variables and can identify causal directions among latent variables, but it still requires that each latent variable set should have at least twice more measured variables as children.

- Moreover, Huang* et al. [2020] considered a special type of confounders in heterogeneous data, where the confounder can be represented as a function of domain index or a smooth function of time, so one may use the known domain index or time index as a surrogate variable to remove the influence from those confounders and thus identify causal structure over observed variables.

- Mixture oracles-based method. Recently, Kivva et al. [2021] proposed a mixture oracles-based method to identify the latent variable graph that allows nonlinear causal relationships. It is based on assumptions that the latent variables are discrete and each latent variable has measured variables as children. Thanks to the discreteness assumption, it can handle more general DAGs over latent variables.

# D Proofs

## D.1 Proof of Theorem 1

**Theorem 1** (Graphical Implication of Rank Constraints in IL$^2$H Graphs). *Suppose $\mathcal{G}$ satisfies an IL$^2$H graph. Under the rank faithfulness assumption, the cross-covariance matrix $\Sigma_{\mathbf{X}_A, \mathbf{X}_B}$ over measured variables $\mathbf{X}_A$ and $\mathbf{X}_B$ in $\mathcal{G}$ (with $|\mathbf{X}_A|, |\mathbf{X}_B| > r$) has rank r, if and only if there exists a subset of latent variables $\mathbf{L}$ with $|\mathbf{L}| = r$ such that $\mathbf{L}$ d-separates $\mathbf{X}_A$ from $\mathbf{X}_B$, and there is no $\mathbf{L}'$ with $|\mathbf{L}'| < |\mathbf{L}|$ that d-separates $\mathbf{X}_A$ from $\mathbf{X}_B$. That is,*

$$rank(\Sigma_{\mathbf{X}_A, \mathbf{X}_B}) = min\{|\mathbf{L}| : \mathbf{L}\ d\text{-separates}\ \mathbf{X}_A\ from\ \mathbf{X}_B\}.$$

*Proof.* Theorem 1 is a special case of Theorem 2.8 in Sullivant et al. [2010] when applied to IL$^2$H graphs. Different from the setting in Sullivant et al. [2010] which access to the full covariance matrix $\Sigma_{\mathbf{V}_{\mathcal{G}}, \mathbf{V}_{\mathcal{G}}}$ is assumed, we only have access to the covariance matrix $\Sigma_{\mathbf{X}_{\mathcal{G}}, \mathbf{X}_{\mathcal{G}}}$ over the measured variables $\mathbf{X}_{\mathcal{G}}$.

It is enough to show that for IL$^2$H graphs, $(C_A, C_B)$ t-separating $\mathbf{X}_A$ from $\mathbf{X}_B$ is equivalent to $\mathbf{L}$ d-separating $\mathbf{X}_A$ from $\mathbf{X}_B$, where $C_A, C_B \subset \mathbf{L}_{\mathcal{G}}$.

Since $\mathbf{L}$ d-separates $\mathbf{X}_A$ from $\mathbf{X}_B$ and since any $X \in \mathbf{X}_{\mathcal{G}}$ cannot be the choke point, we can choose $C_A = \mathbf{L}$ and $C_B = \emptyset$, so that $(C_A, C_B)$ t-separates $\mathbf{X}_A$ from $\mathbf{X}_B$ in IL$^2$H graphs.

Therefore, combining Theorem 2.8 in Sullivant et al. [2010] and the above equivalence, the theorem is proved. □

## D.2 Proof of Theorem 2

**Theorem 2** (Measurement as a surrogate). *Suppose $\mathcal{G}$ is an IL$^2$H graph. Denote by $\mathbf{A}, \mathbf{B} \subseteq \mathbf{V}_{\mathcal{G}}$ two subsets of variables in $\mathcal{G}$, with $\mathbf{A} \cap \mathbf{B} = \emptyset$. Furthermore, denote by $\mathbf{X}_A$ the set of measured variables that are d-separated by $\mathbf{A}$ from all other measures, and by $\mathbf{X}_B$ the set of measured variables that are d-separated by $\mathbf{B}$ from all other measures. Then $\texttt{rank}(\Sigma_{\mathbf{A}, \mathbf{B}}) = \texttt{rank}(\Sigma_{\mathbf{X}_A, \mathbf{X}_B})$.*

*Proof.* According to Theorem 1, the rank of $\Sigma_{X_A, X_B}$ is the minimal number of latent variables $\mathbf{L}$ that d-separate $X_{\mathbf{A}}$ and $X_{\mathbf{B}}$; that is, $\mathbf{L}$ block all paths between $X_{\mathbf{A}}$ and $X_{\mathbf{B}}$ with the smallest cardinality. Furthermore, since $\mathbf{A}$ d-separates $\mathbf{X}_A$ from all other measures, $\mathbf{X}_A$ are the measured pure descendants of $\mathbf{A}$. Similarly, $\mathbf{X}_B$ are the measured pure descendants of $\mathbf{B}$. Moreover, given the structure of IL$^2$H graphs, $\mathbf{L}$ also block all paths between $\mathbf{A}$ and $\mathbf{B}$ with the smallest cardinality. Therefore, the rank of $\Sigma_{\mathbf{A}, \mathbf{B}}$ is also $|\mathbf{L}|$, equivalent to that of $\Sigma_{\mathbf{X}_A, \mathbf{X}_B}$. □

## D.3 Proof of Theorem 3

**Theorem 3** (Correct Cluster Condition). *Suppose $\mathcal{G}$ is an IL$^2$H graph with measured variables $\mathbf{X}_{\mathcal{G}}$. Consider the output $\mathcal{G}'$ from applying findCausalClusters over $\mathbf{X}_{\mathcal{G}}$. If none of the clusters in $\mathcal{G}'$ is the bond set in $\mathcal{G}$, then all latent atomic covers have been correctly identified.*

Before the proof of Theorem 3, we first give the following lemma which shows that if $\mathbf{X} \subseteq \mathbf{X}_{\mathcal{G}}$ is not a correct cluster in the true graph $\mathcal{G}$, then $\mathbf{X}$ forms a bond set. In other words, If a set of measured variables does not form a bond set of $\mathcal{G}$, then it must form a correct cluster.

**Lemma 1** (Fake-Cluster $\Rightarrow$ Bond Set). *A set of measured variables $\mathbf{X} \subseteq \mathbf{X}_{\mathcal{G}}$ that is mistakely tested as a rank-deficient set by findCausalClusters but does not form a cluster in the true graph $\mathcal{G}$ (that is, a fake cluster), only if $\mathbf{X}$ forms a bond set of $\mathcal{G}$.*

Below, we first give the proof of Lemma 1.

*Proof.* Suppose we found a rank deficient set of variables $\mathbf{X} = \{X_1, X_2, \cdots\}$ which is not a cluster in $\mathcal{G}$. This implies that (i) there exists at least two disjoint latent atomic covers, $\mathbf{L}_1, \mathbf{L}_2$, which d-separate variables in $\mathbf{X}$ from all other measures, and (ii) $\mathbf{X}$ does not contain all the pure children of either $\mathbf{L}_1$ nor $\mathbf{L}_2$, because otherwise, due to the IL$^2$H requirement that each $\mathbf{L}_i$ has $> |\mathbf{L}_i|$ pure children, we would have discovered that a subset of $\mathbf{X}$ was rank deficient and clustered together earlier by

*findCausalClusters*. This implies that $\mathbf{L}_1, \mathbf{L}_2$ will d-separate the remaining pure child of $\mathbf{L}_1$ from that of $\mathbf{L}_2$, implying that $\mathbf{X}$ is a bond set. $\qquad\square$

Now we are ready to prove Theorem 3.

*Proof.* We will prove that all latent atomic covers can be correctly identified *from bottom to top*, if there is no bond set.

First, from Lemma 1 we know that if a set of measured variables is not a bond set, then they form a correct causal cluster, and thus the corresponding identified latent cover is correct. Denote the latent atomic covers identified at this step by $\mathbf{L}'_1$, and denote by $\mathcal{G}'_1$ the currently estimated graph.

After identifying the latent atomic covers at the downmost level, we next continue to form causal clusters from root variables in $\mathcal{G}'_1$, including $\mathbf{L}'_1$ and the remaining measured variables that did not form clusters in the previous step. If any of the latent covers in $\mathbf{L}'_1$ have latent children $\mathbf{L}_i$ in the true graph $\mathcal{G}$ that have not been identified in the current step, then reverse these edges such that the children become parents. Such an operation will not affect the discovery of latent covers, and $\mathbf{L}_i$ will be found in later steps. So by further leveraging Lemma 1 and by treating $\mathbf{L}'_1$ and the remaining measured variables that did not form clusters in the previous step as "measured variables", this step results in correctly identified latent covers $\mathbf{L}'_2$ and estimated graph $\mathcal{G}'_2$.

We can now iteratively repeat the previous step to discover new latent atomic covers from the root variables in the estimated graph in the previous step, until no more rank deficient sets can be found. $\qquad\square$

## D.4 Proof of Theorem 4

**Theorem 4** (Correcting Clusters). *Denote by $\mathcal{G}'$ the output from findCausalClusters and by $\mathcal{G}$ the true graph. For a latent atomic cover $\mathbf{L}'$ in $\mathcal{G}'$, if the measured pure descendants of $\mathbf{L}'$ is a bond set in the true graph $\mathcal{G}$, then there exist a set of siblings $\mathbf{S}$ of $\mathbf{L}'$ in $\mathcal{G}'$, a set of children $\mathbf{C}$ of $\mathbf{L}'$, and a set of grandparents $\mathbf{P}$ of $\mathbf{L}'$, such that $\mathcal{M}_{\mathcal{G}'}(\mathbf{S} \cup \mathbf{C} \cup \mathbf{P})$ forms a cluster that is not a bond set in $\mathcal{G}$.*

*Proof.* Suppose the measured pure descendants of $\mathbf{L}'$, $\mathbf{X} := \mathcal{M}_{\mathcal{G}'}(\mathbf{L}')$, is a bond set in the true graph $\mathcal{G}$. Denote by $\mathbf{L}_S \subseteq \mathbf{L}_{\mathcal{G}}$ the minimal set of latent variables in $\mathcal{G}$ that d-separates $\mathbf{X}$ from all other measures $\mathbf{X}' := \mathbf{X}_{\mathcal{G}} \backslash \mathbf{X}$, and since $\mathbf{X}$ is a bond set, $\mathbf{L}_S$ also d-separates some disjoint partition of measures $\mathbf{X}_i \subset \mathbf{X}'$ from $\mathbf{X}_j \subset \mathbf{X}'$, and accordingly, denote by $\mathcal{G}_i$ the subgraph that contains measures $\mathbf{X}_i$ and by $\mathcal{G}_j$ the subgraph that contains measures $\mathbf{X}_j$. Moreover, denote by $\mathbf{L}_i \subset \mathbf{L}_S$ the set of latent variables that d-separates $\mathbf{X}_i$ from other measures. The proof contains three steps.

In step 1, we show that for each $\mathbf{L}'$'s sibling, its measured pure descendants are the same set of measured variables as that in the subgraph $\mathcal{G}_i$ for some $i$. To this end, we first show that variables in each subgraph $\mathcal{G}_i$ only formed clusters with variables in the same $\mathcal{G}_i$. Without loss of generality, suppose there are only two subgraphs. Suppose for contradiction that we discovered a cluster of variables $\mathbf{V}$ such that $\mathcal{M}_{\mathcal{G}'}(\mathbf{V})$ comprises measures from $\mathcal{G}_1, \mathcal{G}_2$. Then $\mathbf{V}$ can be separated into variable sets $\mathbf{V}_1, \mathbf{V}_2$ belonging to $\mathcal{G}_1, \mathcal{G}_2$ respectively. However, the minimal d-separating set for $\mathbf{V}_1, \mathbf{V}_2$ must not overlap, since they are in different subgraphs. This implies that either $\mathbf{V}_1, \mathbf{V}_2$ forms a rank deficient set by itself, which should have been discovered earlier. Hence, we reach a contradiction.

We next show that for each $\mathbf{L}'$'s sibling, its measured pure descendants are not a proper subset of the measured variables as that in $\mathcal{G}_i$. If it was the case, the variables in $\mathcal{G}_i$ would continue to form variables with one another until $\mathbf{L}_i$ is in the minimal d-separating set, where $\mathcal{M}_{\mathcal{G}'}(\mathbf{A}') = \mathbf{X}_i$, hence showing the claim.

In step 2, we show that the cardinality of each sibling of $\mathbf{L}'$ is equal to $|\mathbf{L}_i|$ for some $i$. We first show that for a variable set $\mathbf{V}_i$ such that $\mathcal{M}(\mathbf{V}_i) \subseteq \mathcal{M}(\mathcal{G}_i)$, it is not possible that $|\mathbf{V}_i| < |\mathbf{L}_i|$. This is because if this was the case, it implies that there exists a latent set of smaller cardinality that separates the measured variables in $\mathcal{G}_i$ from the rest of the graph, which contradicts the fact that $\mathbf{L}_i$ is minimal. So it is always the case that $|\mathbf{V_i}| \geq |\mathbf{L}_i|$.

Next, we show that as long as any such variable set $|\mathbf{V}_i| > |\mathbf{L}_i|$, it will be able to form a cover with cardinality $< |\mathbf{V}_i|$. This is because if no rank deficient sets exist for cardinality $k < |\mathbf{L}_i|$,

$\{\mathbf{V}_i$ : the measured variables in $\mathbf{V}_i$ = the measured variables in $\mathcal{G}_i\}$ will form a cluster of cardinality $|\mathbf{L}_i|$, since $|\mathbf{L}| > |\mathbf{L}_i| < |\mathbf{V_i}|$.

Thus, by combining step 1 and step 2, we have shown that each sibling of $\mathbf{L}'$ corresponding to $\mathbf{L}_i$ for some $i$.

Finally, in step 3, we show that, for $\mathbf{L}'$, there exist a set of siblings $\mathbf{S}$, a set of children $\mathbf{C}$, and a set of grandparent $\mathbf{P}$, so that their union $\mathbf{S} \cup \mathbf{C} \cup \mathbf{P}$ will not form a bond cover. Note that the measured pure descendants of siblings or grandparents are $\mathbf{X}_i$, which is the reason why we need to consider the siblings and grandparents. Moreover, note that the reason $\mathbf{L}'$ is formed as bond cover is that when it is formed, its slibings have not been found yet. Intuitively, now its siblings have been found, so we can find the correct clusters.

Suppose for contradiction that refining clusters will discover a new bond cover $\mathbf{L}_{bond}$, and without loss of generality, suppose the minimal d-separating set involves some distinct covers $\mathbf{L}_A, \mathbf{L}_B$. Each of $\mathbf{L}_A, \mathbf{L}_B$ must respectively d-separate some partition of variables $\mathbf{V}_A, \mathbf{V}_B \subset \mathbf{V}$ from all other variables remaining in $\mathbf{V}$. We also know that in order to find rank deficiency, $\|\mathbf{L}_A\| + \|\mathbf{L}_B\| < \|\mathbf{V}_A\| + \|\mathbf{V}_B\|$, implying that $\|\mathbf{L}_A\| < \|\mathbf{V}_A\|$ or $\|\mathbf{L}_B\| < \|\mathbf{V}_B\|$. However, since $\mathbf{L}_A, \mathbf{L}_B$ d-separates $\mathbf{V}_A, \mathbf{V}_B$ from all other variables respectively, testing either $\mathbf{V}_A, \mathbf{V}_B$ must have been rank deficient. Since either of them are over a smaller latent cardinality ($\|\mathbf{L}_A\|, \|\mathbf{L}_B\| < \|\mathbf{L}_A\| + \|\mathbf{L}_B\|$), one of them must have been discovered as a cluster earlier. Hence, we reach a contradiction. $\quad\square$

## D.5 Proof of Lemma 5

**Lemma 5** (Rank Invariance)**.** *Denote by $\mathcal{G}'$ the output from findCausalClusters and by $\mathbf{L}$ a latent atomic cover in $\mathcal{G}'$. Then the rank constraints over $\mathbf{X}_G$ prescribed by $\mathcal{G}'$ before and after the operation makeRoot($\mathbf{L}$) are identical.*

*Proof.* For an $IL^2H$ graph $\mathcal{G}'$ and a latent atomic cover $\mathbf{L}$ in $\mathcal{G}'$, after applying the makeRoot operator to $\mathbf{L}$, which results in $\mathcal{G}''$, $\mathcal{G}'$ and $\mathcal{G}''$ are in the same Markov equivalence class. Therefore, $\mathcal{G}'$ and $\mathcal{G}''$ have the same rank constraints. $\quad\square$

## D.6 Proof of Lemma 6

**Lemma 6** (Cross-Cover Test)**.** *Given a set of variables $\mathcal{S}$, consider two latent atomic covers $\mathbf{L}_A, \mathbf{L}_B \in \mathcal{S}$, and a potential separating set $\mathbf{C} = \{\mathbf{L}_{C_i}\} \subseteq \mathcal{S}\backslash\{\mathbf{L}_A, \mathbf{L}_B\}$. For each $\mathbf{L}_{C_i}$, consider $\mathbf{C}_i^A, \mathbf{C}_i^B \subseteq PCh(\mathbf{L}_{C_i})$ with $\mathbf{C}_i^A, \mathbf{C}_i^B \neq \emptyset$ and $\mathbf{C}_i^A \cap \mathbf{C}_i^B = \emptyset$, and denote the cardinality $k_i^A := min(|\mathbf{L}_{C_i}|, |\mathbf{C}_i^A|)$, $k_i^B := min(|\mathbf{L}_{C_i}|, |\mathbf{C}_i^B|)$, respectively. Then there is no edge between $\mathbf{L}_A$ and $\mathbf{L}_B$ if and only if there exists a separating set $\mathbf{C}$ such that $\texttt{rank}(\Sigma_{\mathcal{A},\mathcal{B}}) < min(|\mathbf{L}_A| + \sum_i k_i^A, |\mathbf{L}_B| + \sum_i k_i^B)$, where $\mathcal{A} = \{\mathbf{L}_A, \mathbf{C}_1^A, \mathbf{C}_2^A, ...\}$ and $\mathcal{B} = \{\mathbf{L}_B, \mathbf{C}_1^B, \mathbf{C}_2^B, ...\}$. In this case, we say that $\mathbf{C}$ satisfies the cross-cover test of $\mathbf{L}_A$ against $\mathbf{L}_B$.*

*Proof.* We first show that if there is no edge between $\mathbf{L}_A$ and $\mathbf{L}_B$, then $\texttt{rank}(\Sigma_{\mathcal{A},\mathcal{B}}) < min(|\mathbf{L}_A| + \sum_i k_i^A, |\mathbf{L}_B| + \sum_i k_i^B)$.

Since there is no edge between $\mathbf{L}_A$ and $\mathbf{L}_B$, there exists a set $\mathbf{C} = \{\mathbf{L}_{C_i}\}$, so that given $\mathbf{C}$, $\mathbf{L}_A$ and $\mathbf{L}_B$ are d-separated. Since $\mathbf{C}_i^A$ and $\mathbf{C}_i^B$ are the children of $\mathbf{L}_{C_i}$, $\mathbf{C}$ d-separates $\mathcal{A}$ from $\mathcal{B}$ as well. Then according to Theorem 7.1, $\texttt{rank}(\Sigma_{\mathcal{A},\mathcal{B}}) = |\mathbf{C}|$. Moreover, since $|\mathbf{C}| < min(|\mathbf{L}_A| + \sum_i k_i^A, |\mathbf{L}_B| + \sum_i k_i^B)$, we have $\texttt{rank}(\Sigma_{\mathcal{A},\mathcal{B}}) < min(|\mathbf{L}_A| + \sum_i k_i^A, |\mathbf{L}_B| + \sum_i k_i^B)$.

Next we show that $\texttt{rank}(\Sigma_{\mathcal{A},\mathcal{B}}) < min(|\mathbf{L}_A| + \sum_i k_i^A, |\mathbf{L}_B| + \sum_i k_i^B)$, then there is no edge between $\mathbf{L}_A$ and $\mathbf{L}_B$.

Now suppose that there are edges between $\mathbf{L}_A$ and $\mathbf{L}_B$. Then $\Sigma_{\mathcal{A},\mathcal{B}}$ is not rank deficient; that is $\texttt{rank}(\Sigma_{\mathcal{A},\mathcal{B}}) = min(|\mathbf{L}_A| + \sum_i k_i^A, |\mathbf{L}_B| + \sum_i k_i^B)$. Therefore, if $\texttt{rank}(\Sigma_{\mathcal{A},\mathcal{B}}) < min(|\mathbf{L}_A| + \sum_i k_i^A, |\mathbf{L}_B| + \sum_i k_i^B)$, then there is no edge between $\mathbf{L}_A$ and $\mathbf{L}_B$. $\quad\square$

## D.7 Proof of Lemma 7

**Lemma 7** (V-Structure Test)**.** *For any unshielded triangle $\mathbf{L}_A - \mathbf{L}_C - \mathbf{L}_B$, let $\mathcal{A}, \mathcal{B}$ be the set of variables in Lemma 6 such that $\Sigma_{\mathcal{A},\mathcal{B}}$ was rank deficient. Let $k = \texttt{rank}(\Sigma_{\mathcal{A},\mathcal{B}})$, $k_1 = \texttt{rank}(\Sigma_{\mathcal{A}\cup\mathbf{L}_C,\mathcal{B}})$, and $k_2 = \texttt{rank}(\Sigma_{\mathcal{A},\mathcal{B}\cup\mathbf{L}_C})$. Then, $\mathbf{L}_A \rightarrow \mathbf{L}_B \leftarrow \mathbf{L}_C$ if and only if $k < \min(k_1, k_2)$.*

*Proof.* We first show that if $\mathbf{L}_A \to \mathbf{L}_C \leftarrow \mathbf{L}_B$, then $k < \min(k_1, k_2)$.

Since $\mathbf{L}_A \to \mathbf{L}_C \leftarrow \mathbf{L}_B$, $\mathbf{L}_C$ cannot be in the separation set of $\mathbf{L}_A$ and $\mathbf{L}_B$; that is, given $\mathbf{L}_C$, $\mathbf{L}_A$ and $\mathbf{L}_B$ are d-connected. Hence, $k < k_1$ and $k < k_2$, and thus $k < \min(k_1, k_2)$.

Next we show that if $k < \min(k_1, k_2)$, then $\mathbf{L}_A \to \mathbf{L}_C \leftarrow \mathbf{L}_B$.

Suppose $\mathbf{L}_A, \mathbf{L}_C, \mathbf{L}_B$ do not form a v-structure; that is $\mathbf{L}_A \to \mathbf{L}_C \to \mathbf{L}_B$ or $\mathbf{L}_A \leftarrow \mathbf{L}_C \leftarrow \mathbf{L}_B$. Then $k = \min(k_1, k_2)$, since $\mathbf{L}_C$ has been considered before in order to achieve rank deficiency of $\Sigma_{\mathcal{A}, \mathcal{B}}$. Therefore, if $k < \min(k_1, k_2)$, then $\mathbf{L}_A \to \mathbf{L}_C \leftarrow \mathbf{L}_B$. $\qquad\square$

## D.8 Proof of Lemma 8

**Lemma 8.** *Suppose $\mathcal{G}$ is an $IL^2H$ graph. The rank constraints are invariant with the minimal-graph operator and the skeleton operator; that is, $\mathcal{G}$ and $O_{skeleton}(O_{min}(\mathcal{G}))$ are rank equivalent.*

*Proof.* We first show that the minimal-graph operator will not change rank deficiency constraints. Denote by $\mathcal{G}_1$ and $\mathcal{G}_2$ the graph before and after applying the minimal-graph operator, respectively. For every latent atomic cover $\mathbf{L}$ in $\mathcal{G}_1$, since those three conditions hold, for any $\mathbf{C} \subseteq PCh_{\mathcal{G}'}(\mathbf{L}')$ and for any $\mathbf{S} \subseteq Sib_{\mathcal{G}'}(\mathbf{L}')$ with $\mathbf{C}, \mathbf{S} \neq \emptyset$, $\mathrm{rank}(\Sigma_{\mathcal{A}, \mathcal{B}}) = |\mathbf{P}|$, where $\mathcal{A} = \mathbf{C} \cup \mathbf{S}$ and $\mathcal{B} = \mathbf{X}_{\mathcal{G}'} \backslash \mathcal{M}_{\mathcal{G}'}(\mathcal{A})$. So, after merging $\mathbf{L}$ to its parents $\mathbf{P}$, the cardinality of the d-separation set between any two sets of variables does not change. Thus, according to Theorem 1, the rank constraints will not change after merging $\mathbf{L}$ to its parents $\mathbf{P}$.

Moreover, it is trivial to show that the skeleton operator will not change rank deficiency constraints, because the d-separation set between any two sets of variables will not change. $\qquad\square$

## D.9 Proof of Theorem 9

**Theorem 9.** *Suppose $\mathcal{G}$ is an $IL^2H$ graph with measured variables $\mathbf{X}_{\mathcal{G}}$. Phases I-II in Algorithm 1 over $\mathbf{X}_{\mathcal{G}}$ can asymptotically identify the latent atomic covers of $O_{min}(\mathcal{G})$, with the first two conditions in Condition 1.*

*Proof.* Theorem 3 has shown that *findCausalClusters* gives correct latent covers when there is no bond set. Furthermore, Theorem 4 shows that even in the presence of bond sets, refining the set over $Ch(\mathbf{L}') \cup Sib(\mathbf{L}') \cup Gp(\mathbf{L}')$, where $\mathcal{M}_{\mathcal{G}'}(\mathbf{L}')$ forms a bond set, can correct the clusters.

Phase II *refineClusters* refines clusters over $Ch(\mathbf{L}') \cup Sib(\mathbf{L}') \cup Gp(\mathbf{L}')$ in a breadth-first search from the root variable, and it ends after refining every latent cover in $\mathcal{G}'$. Therefore, with this refining procedure, we will derive correct latent covers. $\qquad\square$

## D.10 Proof of Theorem 10

**Theorem 10.** *Suppose $\mathcal{G}$ is an $IL^2H$ graph with measured variables $\mathbf{X}_{\mathcal{G}}$. Algorithm 1, including Phases I-III, over $\mathbf{X}_{\mathcal{G}}$ can asymptotically identify the Markov equivalence class of $O_{min}(O_s(\mathcal{G}))$.*

*Proof.* Theorem 5 has shown that Phases I-II can find the correct clusters and latent atomic covers of $O_{min}(\mathcal{G})$. Moreover, Lemma 2 and Lemma 3 have shown that by performing Cross-Cover Test and V-Structure Test, the skeleton and v structure among every triple of latent variables can be correctly identified.

Phase III *refineEdges* refines the edges over $\mathbf{L}' \cup PCh(\mathbf{L}') \cup PCh(PCh(\mathbf{L}'))$ by performing Cross-Cover Test and V-Structure Test in a depth-first search from the root variable, and it ends until $\mathbf{L}'$ does not have latent children. Therefore, with this refining procedure, we will derive correct skeletons and v structures. $\qquad\square$

# E  More Explanations on Identifiability Conditions, Algorithms, and Theorems

## E.1  More Explanations on Definition 3 (*Effective Cardinality*)

The effective cardinality, defined in Definition 3, can be estimated with the following procedure.

    $j \leftarrow 1$;
    $\mathbf{C} \leftarrow PCh_{\mathcal{G}}(\mathbf{L})$;
    **while** $j < |\mathbf{L}|$ **do**
        Find the largest subset of variables $\mathbf{C}' \subseteq \mathbf{C}$ such that $|\mathbf{C}'| > |Pa_{\mathcal{G}}(\mathbf{C}')| = j$;
        Introduce a set of latents $\mathbf{L}'$ with $|\mathbf{L}'| = |Pa_{\mathcal{G}}(\mathbf{C}')|$;
        add $\mathbf{L}'$ as new children of $Pa_{\mathcal{G}}(\mathbf{C}')$;
        $\mathbf{C} \leftarrow \mathbf{C} \backslash \mathbf{C}' \bigcup \mathbf{L}'$;
        $j$ += 1;
    **end while**
    *return* $|\mathbf{C}|$

For example, for Figure 1(a), the effective cardinality of the pure children of $\{L_7, L_8\}$ is 3, because $|\{X_4, X_5\}| > |\{L_7\}|$ and we replace $\{X_4, X_5\}$ with a single latent variable $L'$, so the cardinality of the resulting children set is $|\{X_6, X_7, L'\}| = 3$.

## E.2  More Explanations on Definition 4 (*Latent Atomic Cover*)

The first two conditions in Definition 4 ensure that there are enough variables in the current active variable set to find the rank deficiency, so that we can determine the latent atomic cover with size $k$. However, note that they may not be the necessary conditions. For example, for the graphs in Figure 8, some of the latent atomic covers only have $k$ neighbors (except for the $k + 1$ pure children), but they are still identifiable.

The first half of the third condition, "there does not exist a partition of $\mathbf{L} = \mathbf{L}_1 \cup \mathbf{L}_2$, so that both $\mathbf{L}_1, \mathbf{L}_2$ satisfy conditions 1 and 2", ensure that the latent atomic cover $\mathbf{L}$ is atomic.

The second half of the third condition, "there does not exist a partition of $\mathbf{L} = \mathbf{L}_1 \cup \mathbf{L}_2$, so that $\{PCh_{\mathcal{G}}(\mathbf{L}_1) \cup PCh_{\mathcal{G}}(\mathbf{L}_2)\} \backslash \mathbf{L} = PCh_{\mathcal{G}}(\mathbf{L})$", is needed in the overlapping case. Consider the following graph: $L_1 \rightarrow \{X_1, X_2, X_3\}, L_2 \rightarrow \{X_3, X_4, X_5\}$. Here, $\mathbf{L}_1 = \{L_1\}$, $\mathbf{L}_2 = \{L_2\}$, and $\mathbf{L} = \{L_1, L_2\}$ are latent atomic covers; however, $\{PCh_{\mathcal{G}}(\mathbf{L}_1) = \{X_1, X_2\}$, $\{PCh_{\mathcal{G}}(\mathbf{L}_2) = \{X_4, X_5\}$, and $\{PCh_{\mathcal{G}}(\mathbf{L}) = \{X_1, X_2, X_3, X_4, X_5\}$, so $\{PCh_{\mathcal{G}}(\mathbf{L}_1) \cup PCh_{\mathcal{G}}(\mathbf{L}_2)\} \backslash \mathbf{L} \neq PCh_{\mathcal{G}}(\mathbf{L})$. Therefore, although both $\mathbf{L}_1$ and $\mathbf{L}_2$ satisfy the first two conditions, $\mathbf{L}$ satisfies Definition 4.

## E.3  More Explanations on Condition 1 (*$IL^2H$ graph*)

The first two conditions in Condition 1 guarantee to identify the latent atomic covers, while the last one is used to identify the edges among latent atomic covers when performing Cross-Cover Test and V-Structure Test.

Specifically, the second condition says that two latent atomic covers that are partly overlapped is not allowed, except for the case that one latent atomic cover is contained in another one; otherwise, the cover creation rule can be non-trivial. For example, for the graph in Figure 9(a), although $PDe_{\mathcal{G}}(\{L_3, L_4\}) \bigcap PDe_{\mathcal{G}}(\{L_4\}) = \{X_3, X_4\}$, it satisfies the second condition in Condition 1 because $\{L_3\} \subset \{L_2, L_3\}$. In contrast, for the graph in Figure 9(b), $PDe_{\mathcal{G}}(\{L_3, L_4\}) \bigcap PDe_{\mathcal{G}}(\{L_4, L_5\}) = \{X_3\}$, but $\{L_3, L_4\}, \{L_4, L_5\}$ are not subsets or descendants of one another. Hence, it does not satisfy the second condition of an $IL^2H$ graph.

## E.4  More Explanations on Algorithm 2 (*findCausalClusters*)

The search procedure in Algorithm 2 (*findCausalClusters*) contains the following two key updates.

- *Latent-atomic-cover size update.* We start to identify latent atomic covers with size $k = 1$. If a rank-deficiency set is not found, then increment $k = k + 1$; otherwise, reset $k = 1$.
- *Active variable set update.* The set of active variables $\mathcal{S}$ is set to $\mathbf{X}_{\mathcal{G}}$ initially. We consider any subset of the latent atomic covers in $\mathcal{S}$ and replace them with their pure children, resulting in $\tilde{\mathcal{S}}$. At

Figure 8: Latent hierarchical graphs where some latent atomic covers only have $k$ neighbors, except for the $k + 1$ pure children, but the graph structure is still identifiable. Note that in many cases, $k$-neighbor is enough; the condition "$k + 1$ neighbors" is sufficient, but not necessary.

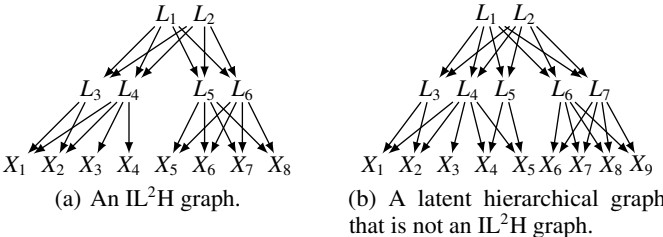

(a) An IL$^2$H graph.

(b) A latent hierarchical graph that is not an IL$^2$H graph.

Figure 9: Examples and counter-examples of IL$^2$H graphs.

the same time, we search for the rank-deficiency set **A** with rank $k$ in $\tilde{\mathcal{S}}$; if it is found, then assign a latent atomic cover **L** of size $k$ as the parent **A**, and accordingly, the active variable set is updated as $(\mathcal{S} \backslash \mathbf{A}) \cup \mathbf{L}$.

Note that in Algorithm 2 *findCausalClusters*, we search over $\tilde{\mathcal{S}}$, instead of $\mathcal{S}$, to avoid adding a certain type of redundant latent variables when some latent atomic covers have overlapping variables, including the v structure; see Figure 10 for an illustration.

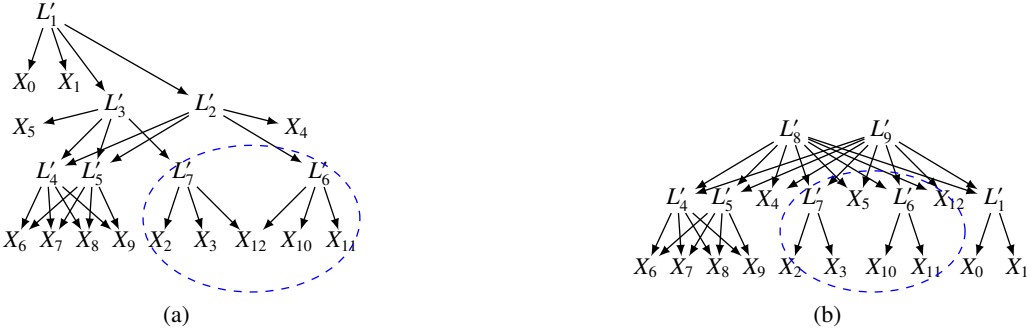

(a)                                                (b)

Figure 10: (a) Output from Phase I *findCausalClusters* with ground truth graph in Figure 7(b), where we consider the children of the active variable set $\mathcal{S}$ (i.e., $\tilde{\mathcal{S}}$) to search for the rank deficiency set. (b) The output graph if directly searching over $\mathcal{S}$ in Phase I; that is, without considering line 4 in Algorithm 2. The output graph without considering $\tilde{\mathcal{S}}$ is not correct; for instance, $X_{12}$ will not be considered as the children of $\{L_6, L_7\}$.

Moreover, in Algorithm 2 *findCausalClusters*, if there are conflicts when the search goes on, then we just ignore it, and such conflicts will be handled in Algorithm 3 *refineClusters*. Also, note that except for the v structure where the measured variable is a collider, other "v structures" in the intermediate output in Algorithms 2 and 3 are not true v structures.

For the illustration of Algorithm 2 given in Figure 2, here we give more detailed explanations. Specifically, we first set $k = 1$ and the active set is $\mathcal{S} = \{X_1, \cdots, X_{16}\}$ and $\tilde{\mathcal{S}} = \mathcal{S}$, and we can find the clusters in (a), and no further cluster can be found with $k = 1$. Then we increase $k$ to 2 with the active set $\mathcal{S} = \{\{L_6\}, \{L_7\}, X_6, \cdots, X_{16}\}$ and $\tilde{\mathcal{S}} = \mathcal{S}$, and then we can find the clusters in (b). Then, the active set is $\mathcal{S} = \{\{L_4, L_5\}, \{L_6\}, \{L_7, L_8\}, \{L_9, L_{10}\}\}$ and we set back $k = 1$, and when $\tilde{\mathcal{S}} = \{\{L_4, L_5\}, X_1, \cdots, X_{11}\}$ we find the cluster in (c). Note that when testing the rank over $\{L_4, L_5\}$ against other variables, we use their measured pure descendants in the currently estimated graph instead. The above procedure is repeated to further find the cluster in (d). Finally, when there are no enough variables for testing, we connect the elements in the active variable set: connecting $\{L_2, L_3\}$ to $\{L_7, L_8\}$ in (e).

### E.5  Explanation of Minimal-Graph Operator and Skeleton Operator

The minimal-graph operator and the skeleton operator will not change the rank constraints, or in other words, graphs before and after applying the operators are indistinguishable with rank constraints.

Give an IL$^2$H graph in Figure 11(a), after applying the minimal-graph operator, the latent atomic cover $L_4$ will be merged to its parent $L_5$, resulting in the graph in Figure 11(b), while the rank constraints will not change. Furthermore, after applying the skeleton operator to the graph in Figure 11(b), $L_1$ has an edge to $X_7$ and $L_3$ has an edge to $X_1$, resulting in the graph in Figure 11(c), which also does not change the rank constraints.

### E.6  Complexity of Algorithm 1

The time complexity of the algorithm is upper bounded by $O(r \sum_{k=2}^{l+1} \binom{m}{k})$, and this bound is further upper bounded by $O(r(1 + m)^{l+1})$, where $m$ is the number of measured variables, $l$ is the cardinality of the largest latent cover of the estimated graph, with $l \ll m$, and $r$ is the number of levels of the estimated hierarchical graph, with $r \ll m$.

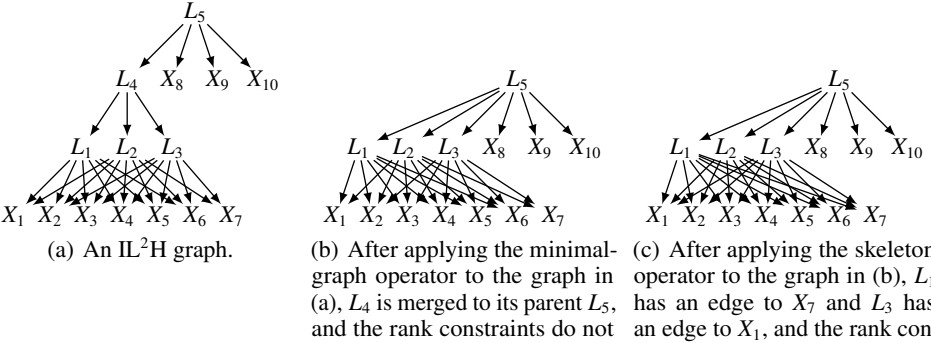

(a) An IL²H graph.

(b) After applying the minimal-graph operator to the graph in (a), $L_4$ is merged to its parent $L_5$, and the rank constraints do not change.

(c) After applying the skeleton operator to the graph in (b), $L_1$ has an edge to $X_7$ and $L_3$ has an edge to $X_1$, and the rank constraints do not change.

Figure 11: Examples of applying the minimal-graph operator and the skeleton operator to an IL²H graph.

### E.7 Illustrative Examples of the Entire Algorithm

Figure 12 and Figure 13 give two illustrative examples of the entire algorithm, showing how each step proceeds.

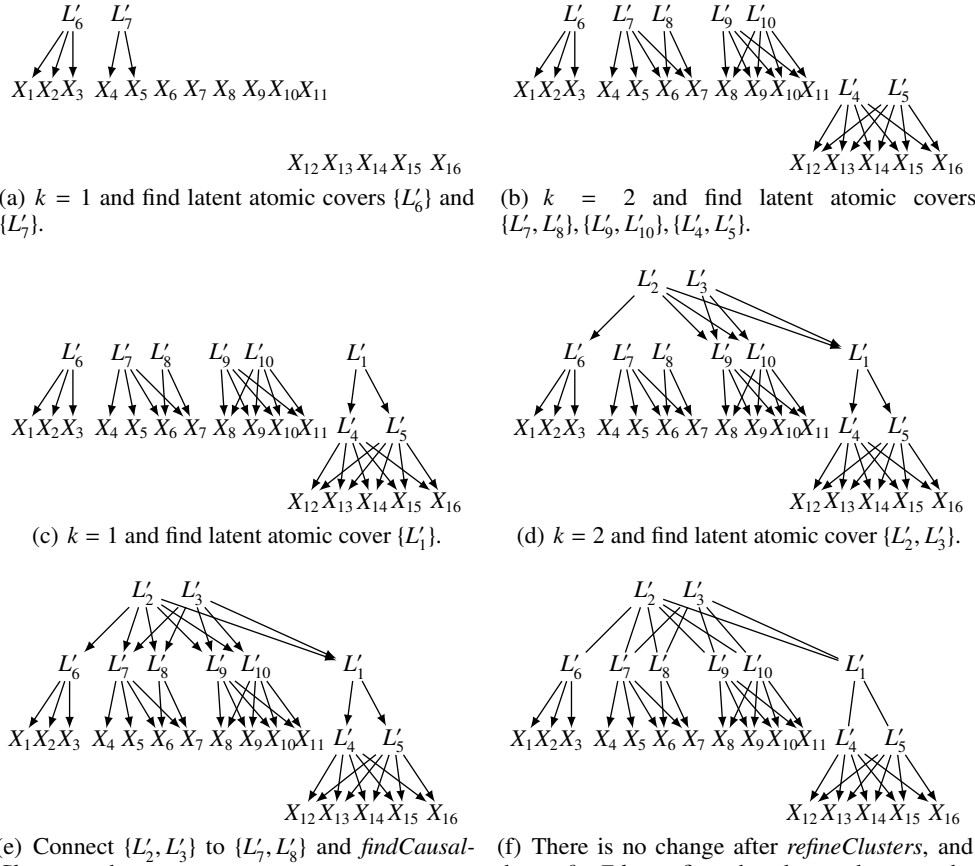

(a) $k = 1$ and find latent atomic covers $\{L_6'\}$ and $\{L_7'\}$.

(b) $k = 2$ and find latent atomic covers $\{L_7', L_8'\}, \{L_9', L_{10}'\}, \{L_4', L_5'\}$.

(c) $k = 1$ and find latent atomic cover $\{L_1'\}$.

(d) $k = 2$ and find latent atomic cover $\{L_2', L_3'\}$.

(e) Connect $\{L_2', L_3'\}$ to $\{L_7', L_8'\}$ and *findCausalClusters* ends.

(f) There is no change after *refineClusters*, and then *refineEdges* refines the edges and outputs the Markov equivalence class.

Figure 12: An illustrative example by applying Algorithm 1 to the measured variables in Figure 7(a).

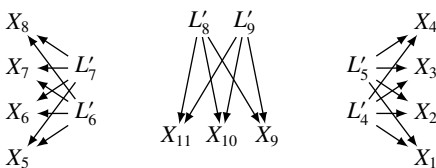

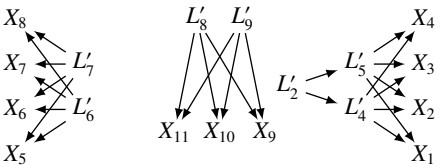

(a) $k = 2$ and find latent atomic covers $\{L'_6, L'_7\}$, $\{L'_4, L'_5\}$, and $\{L'_8, L'_9\}$.

(b) $k = 1$ and find a latent atomic cover $\{L'_2\}$.

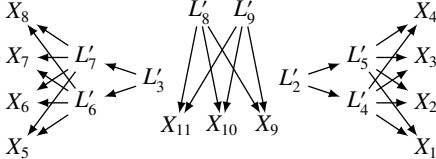

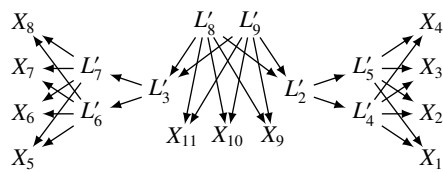

(c) $k = 1$ and find a latent atomic cover $\{L'_3\}$.

(d) Connect $\{L'_8, L'_9\}$ to $\{L'_2\}$ and to $\{L'_3\}$, and *findCausalClusters* ends.

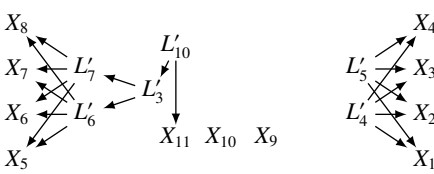

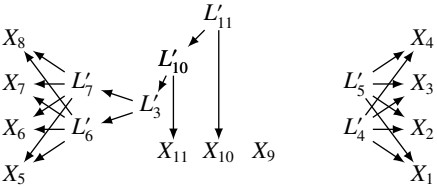

(e) Refine $\{L'_2\}$ by first removing $\{L'_2\}$ and its parents $\{L'_8, L'_9\}$ and performing *findCausalClusters* over $\{L'_3, L'_4, L'_5, X_9, X_{10}, X_{11}\}$, and then we can find a latent cover $\{L'_{10}\}$.

(f) Next perform *findCausalClusters* over $\{L'_{10}, L'_4, L'_5, X_9, X_{10}\}$ we can find a latent cover $\{L'_{11}\}$.

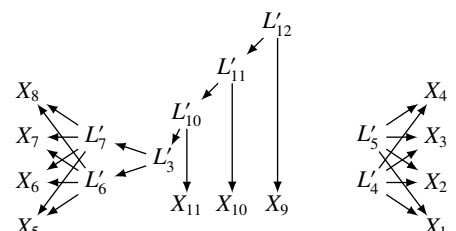

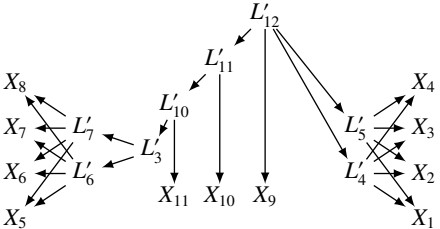

(g) Next perform *findCausalClusters* over $\{L'_{11}, L'_4, L'_5, X_9\}$ we can find a latent cover $L'_{12}$.

(h) Connect $\{L'_{12}\}$ to $\{L'_4, L'_5\}$.

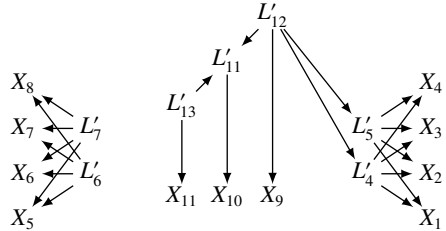

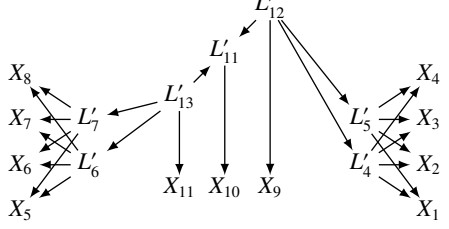

(i) Refine $\{L'_3\}$ by first removing $\{L'_3\}$ and its parents $\{L'_{10}\}$ and performing *findCausalClusters* over $\{L'_6, L'_7, L'_{11}, X_{11}\}$, and then we can find a latent cover $\{L'_{13}\}$.

(j) Connect $\{L'_{13}\}$ to $\{L'_6, L'_7\}$.

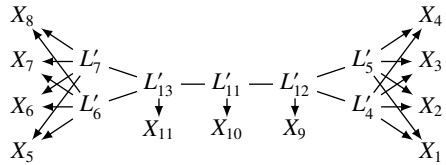

(k) Perform *refineEdges* to refine the edges and output the Markov equivalence class.

Figure 13: An illustrative example by applying Algorithm 1 to the measured variables in Figure 8(c).