# OpenReview forum: "Latent Hierarchical Causal Structure Discovery with Rank Constraints"
_NeurIPS.cc/2022/Conference — NeurIPS 2022 Accept_

### Official Review · Reviewer_ZSqv · 2022-07-05

**Rating:** 8
**Confidence:** 4
**Soundness:** 3 good
**Presentation:** 3 good
**Contribution:** 3 good

**Summary:**

## Summary of paper:

This work deals with the task of causal discovery of linear additive models in the presense of latent (hidden) variables. Under various assumptions on the graph structure, this work proposes an algorithm to recover the Markov equivalence class of the underlying graph and they show that the recovered graph is asymptotically correct up to certain graph theoretic operators. Most prior works either assume the absence of latent variables (which almost always fails in real-life) or do not necessarily recover the causal structure among the latent variables. It's worth noting that few works have also attempted the latter difficult task of recovering the latent causal structure -- for the linear case, see [1], [2] and for the nonlinear case, see [3].

This work deals with the linear case. The basic strategy is to use rank constraints (which can be exploited because of the linearity assumption) to build the latent graph. Such approaches have been utilized before in other works, e.g. [4], [5]. The algorithm has 3 parts - (a) find "atomic" clusters of latent variables using a greedy approach; (b) fixing incorrect clusters that could be formed; and finally, (c) fix the local structure using the proposed cross-cover test and v-structure test. The overall algorithm proposed here seems similar in spirit to the work of [2] but is technically more involved. With this algorithm, they are able to recover a latent causal graph and show that in the infinite data limit, the graph they recover is in the same equivalence class as the true graph, up to the "minimal graph" and "skeleton" operators (definitions 9-10).

### References:

[1] Animashree Anandkumar, Daniel Hsu, Adel Javanmard, and Sham Kakade. Learning linear bayesian networks with latent variables. In International Conference on Machine Learning, pages 249–257, 2013

[2] Feng Xie, Ruichu Cai, Biwei Huang, Clark Glymour, Zhifeng Hao, and Kun Zhang. Generalized independent noise condition for estimating latent variable causal graphs. In Advances in Neural Information Processing Systems, pages 14891–14902, 2020

[3] Bohdan Kivva, Goutham Rajendran, Pradeep Ravikumar, and Bryon Aragam. Learning latent causal graphs via mixture oracles. Advances in Neural Information Processing Systems, 34, 2021

[4] Ricardo Silva, Richard Scheine, Clark Glymour, and Peter Spirtes. Learning the structure of linear latent variable models. Journal of Machine Learning Research, 7(Feb):191–246, 2006

[5] Erich Kummerfeld and Joseph Ramsey. Causal clustering for 1-factor measurement models. In Proceedings of the 22nd ACM SIGKDD International Conference on Knowledge Discovery and Data Mining, pages 1655–1664. ACM, 2016.

**Questions:**

Main questions are raised in the strengths and weaknesses part above.

## Other questions/comments:
1. In L98, what is meant by L and pure children being fully connected? Every pair of vertices from within L, within its pure children and across L and its pure children are all connected by an edge?
2. What is the objective of appendix G.1? It doesn't seem to explain Definition 3 as the algorithm in G.1 seems to modify the graph (esp. L287-288). But Definition 3 is for a static graph.
3. In theorem 7, what is meant by a R-2H graph?
4. Theorem 7 says that the correct Markov equivalence class is learned asymptotically. In the proof of theorem 7, it'd be good to clarify why the algorithm works in the limit. In other words, it's not clear where the asymptotic is coming from.
5. How are the latent graphs constructed in the synthetic examples? Moreover, the code is not released so it's impossible to figure out how the synthetic examples were generated.
6. What is the running time of the algorithm? It seems like Algorithm 2 for phase 1 seems to take exponential time in the input size but this should be clarified.
7. The work [2] seems most closely related to this work. There is a comparison to it in Appendix E. The way it's stated, the main difference seems technical - reducting the number of observed variables needed for latent variables. Is there some other conceptual difference between the works?
8. The caption for figure 12 is written over the page number in the appendix.

**Limitations:**

Limitations of the theory are addressed and the authors state there are no potential negative societal impacts. I tend to agree as this is predominantly a theoretical paper.

**Strengths And Weaknesses:**

## Summary of evaluation

The paper obtains nontrivial and potentially useful results. However, the paper is poorly written, with some misleading remarks and definitions. Moreover, the main condition is a bit hard to interpret without diving deep into the proofs and also, there is insufficient comparison to [3], which I think is doing a fairly more complicated setting than this one (although with different assumptions). See the weaknesses section for more details. Despite this, I appreciate the difficulty surpassed in this work, which I believe is inherent for this difficult task of latent causal discovery, that is slowly gaining traction in the community.

## Strengths:
1. Although the work builds on various known ideas such as exploiting rank deficiency and identifying v-structures, there is significant technical novelty in this work such as the 3-step algorithm that gradually builds the latent graph.
2. The guarantees improve on some prior works such as [1], [2] on a technical level by weakening the required number of observed children for latent variables. That said, other assumptions are made especially regarding the graph structure.
3. The algorithms are clear to follow and are illustrated with examples (see appendix G and H)

## Weaknesses:
1. It is claimed in the introduction that previous works do not handle hierarchical structures (also L66-67 in the appendix). Unfortunately, this doesn't seem right because both [2], [3] that are cited here do seem to handle such structures (see figure 1 in either paper). Moreover, these works handle far more complicated latent structures than the ones considered here (see the point below). Could the authors comment on this?
2. Based on my understanding, the first and foremost assumption in this work is that the graph has a hierarchical structure. Note very importantly that this is a **strict subset** of directed acyclic graphs (DAGs). Therefore, this work does not handle general DAGs. This assumption should be explicitly stated and clarified.
3. In the entire abstract and introduction, it is not mentioned that this paper is dealing with a **linear** model. It only shows up in the formal statement in Section 2. However, this is a highly crucial assumption that *significantly* simplifies the task of causal discovery. For instance, say we want to handle nonlinear models  using the same idea. A first stumbling block in this general setting is whether identifying rank-deficiencies, the engine of this paper, can be exploited again.
4. The first definition Definition 2 is confusing. The phrase "Variables V are pure children of variables L" is defined. However, at this point, I don't see why V is uniquely defined, there could be multiple such Vs. Therefore the next sentence "we denote the pure children of L by PCh_G(L)" does not make sense. Could the authors clarify this?
5. More concretely, it's not clear how the Pure Children definition works if L has connections within itself. For example, take figure 1(a) and assume that the edge L1->L3 is not present for now. Suppose L = {L_1, L_3}. Then PCh_G(L) = {L2, L7, L8, L9}. At this point, since L2 and L7 are connected, I don't understand what PCh_G(PCh_G(L)) is. Because of this, how do we define pure descendants of L?
6. The assumptions of condition 1 (which builds on definition 4) seem hard to interpret. I may be missing something here, but based on my understanding, Appendix G.2, G.3 (in particular L297-298, L311-313) seems to explain how they are relevant *for the proposed algorithm to work*, but not as to why they are *reasonable* assumptions to make. It would be good if the authors could clarify this.
7. Building on the above point, these graph theoretic assumptions are stated to be mild. Why are they mild assumptions?
8. A proper comparison to [3] in appendix E should be added. As stated earlier, it seems like [3] handle general DAGs, not just hierarchical DAGs. Moreover, they handle general non-linear causal relationships instead of just linear and also report the more natural SHD metrics (see below). So its important to clarify which fiber of research this line of work develops on a conceptual level.
9. The reported error metric is the percentage of correctly identified causal clusters over measured variables. Is there a reason a standard metric such as Structural Hamming Distance is not reported? Although I agree with L82-83 in that measuring accuracy is tricky for this problem, other works such as [3] have reported the SHD metric.

---

> ### Author Response · Authors · 2022-08-02
> **Responses to Reviewer ZSqv (part 3/3)**
>
> **Q15**: ``What is the running time of the algorithm? It seems like Algorithm 2 for phase 1 seems to take exponential time in the input size but this should be clarified"
>
> **A15**: Thanks for the comments. The complexity of the algorithm is upper bounded by $O(r \sum_{k=2}^{l+1}\binom{n}{k})$, and this bound is further upper bounded by $O(r(1+n)^{l+1})$, where $n$ is the number of measured variables, and $l$ is the cardinality of the largest latent cover of the estimated graph, with $l \ll n$, and $r$ is the number of layers of the estimated hierarchical graph, with $r \ll n$. We have included the analysis in lines 57-60 in Appendix B on the revision.
>
> Moreover, for the tested graph Figure 7(a) in our experiments, it takes around 40 seconds when the sample size is 2000, running on MacBook Pro M1.
>
> **Q16**: ``The work [2] seems most closely related to this work. There is a comparison to it in Appendix E. The way it's stated, the main difference seems technical - reducing the number of observed variables needed for latent variables. Is there some other conceptual difference between the works?"
>
> **A16**: We would like to mention that there are three main differences: (1) The work [2] requires that all noise terms follow non-Gaussian distributions, while our work does not. (2) The work [2] requires that all latent variables have measured variables as children, while in our work, the children of a latent variable can be either latent or measured. (3) In [2], for each latent variable set with size $k$, it should have at least $2k$ **measured** pure children, while our work only needs $k+1$ pure children, with much weaker structural constraints. For instance, [2] clearly cannot handle the cases in Figures 1, 5, and 7 in our paper.  We have included this discussion in Appendix F.
>
> **Q17**: ``The caption for figure 12 is written over the page number in the appendix"
>
> **A17**: Thanks for your careful reading. It has been fixed.

---

> > ### Comment · Reviewer_ZSqv · 2022-08-06
> > **Reviewer response to rebuttal**
> >
> > Thanks for the detailed reponse. My concerns have been addressed, so I'm raising my score. Many of the comparisons to prior works are fairly technical but after a detailed understanding, I stand by my original comment that the difficulty surpassed in this work is nontrivial. With the improved presentation, I'm recommending acceptance.

---

> > > ### Author Response · Authors · 2022-08-06
> > > **Thank you very much for checking the response and updated paper and updating your recommendation**
> > >
> > > We appreciate you checking our response and the updated paper carefully. We are very happy that you found the response and update paper helpful to address your concerns. Your valuable comments have helped improve our presentation a lot. Thank you very much!

---

> ### Author Response · Authors · 2022-08-02
> **Responses to Reviewer ZSqv (part 2/3)**
>
> **Q6**: “The assumptions of condition 1 seem hard to interpret... Appendix G.2, G.3 seems to explain how they are relevant for the proposed algorithm to work, but not as to why they are reasonable assumptions to make. It would be good if the authors could clarify this.”
>
> **A6**: Thank you for the thoughts. Condition 1 is mainly about an assumption on the number of neighbors, including the number of pure children (to guarantee there is sufficient information to determine the relevant latent variables), and these structural constraints are much weaker than the previous approaches that handle continuous variables, including references [1,2,4,5]. Furthermore, note that [3] is designed for discrete latent variables, where the discreteness assumption helps the identification.
>
> **Q7**: "Building on the above point, these graph theoretic assumptions are stated to be mild. Why are they mild assumptions?"
>
> **A7**: The assumptions are clearly milder compared to the assumptions in previous methods that also consider continuous data. For instance, compared to latent tree models, our model allows multiple paths between every pair of variables and allows each variable to have multiple parents. Compared to the methods in the reference [1,2,4,5], our model allows the case where some latent variables do not have measured variables as children. Moreover, compared to [1,2,4,5], the requirement on the number of pure children that each latent variable set has is much weaker.
>
> **Q8**: “A proper comparison to [3] in appendix E should be added. As stated earlier, it seems like [3] handle general DAGs, not just hierarchical DAGs. Moreover, they handle general non-linear causal relationships instead of just linear. So its important to clarify which fiber of research this line of work develops on a conceptual level.”
>
> **A8**: Following your suggestion, a comparison to [3] has been added to Appendix F. Note [3] requires that all latent variables be **discrete** and must have at least one measured variables as children, while our model is designed for continuous variables. The discreteness assumption benefits the identification of more general DAGs. Our work mainly follows the research line by [1,2,4,5], where we all work on linear continuous data, and moreover, note that the hierarchical DAG in our paper is more general than the measurement models considered in [1,2,4,5].
>
> **Q9**: “Is there a reason a standard metric such as Structural Hamming Distance is not reported? Although I agree with L82-83 in that measuring accuracy is tricky for this problem, other works such as [3] have reported the SHD metric.”
>
> **A9**:Thanks for your suggestion. The main reason why we did not use SHD is that it is hard to construct a mapping between the estimated latent variables and the ground-truth latent variables, especially for the case where latent variables do not have measured variables as children (note that [3] does not consider such cases).
>
> **Q10**: ``In L98, what is meant by L and pure children being fully connected? Every pair of vertices from within L, within its pure children and across L and its pure children are all connected by an edge?"
>
> **A10**: The answer to the last question is no. Denote by C the pure children of L. “...fully connected” means that for any pair $(L_i, C_j)$, with $L_i \in L$ and $C_j \in C$, there is an edge from $L_i$ to $C_j$.
>
> **Q11**: ``What is the objective of appendix G.1? It doesn't seem to explain Definition 3 as the algorithm in G.1 seems to modify the graph (esp. L287-288). But Definition 3 is for a static graph"
>
> **A11**: Appendix G1 provides an efficient procedure to determine the effective cardinality in a recursive way, based on Definition 3.
>
> **Q12**: ``In theorem 7, what is meant by a R-2H graph?"
>
> **A12**: Thanks for pointing out the typo (sorry about it). It should be IL$^2$H. Has been corrected.
>
> **Q13**: ``Theorem 7 says that the correct Markov equivalence class is learned asymptotically. In the proof of theorem 7, it'd be good to clarify why the algorithm works in the limit. In other words, it's not clear where the asymptotic is coming from"
>
> **A13**: The asymptotic results come from the correctness of the rank tests, which our algorithm depends on. More details on rank tests were given in Appendix C.
>
> **Q14**: ``How are the latent graphs constructed in the synthetic examples? Moreover, the code is not released so it's impossible to figure out how the synthetic examples were generated"
>
> **A14**: The graphs in synthetic experiments were given in Figures 5-7.

---

> ### Author Response · Authors · 2022-08-02
> **Responses to Reviewer ZSqv (part 1/3)**
>
> We truly appreciate your time dedicated to reviewing this paper and your insightful and encouraging comments. Below, please see our responses, as well as clarifications on the concept of ``hierarchical structures” and the differences to previous works, especially to [2] and [3].
>
> **Q1**: “It is claimed in the introduction that previous works do not handle hierarchical structures...this doesn't seem right because both [2], [3] that are cited here do seem to handle such structures (see figure 1 in either paper)"
>
> **A1**: We would like to clarify on the concept of ``latent hierarchical structures”, which means that the children of latent variables can either be latent or measured; a special case is that some latent variables do not have measured children, such as that in Figure 1(a). However, [2] requires each latent variable set has twice **measured** variables as children (see its Definition 1). [3] also requires each latent variable has **measured** variables as children (see its Assumption 3.1), because, otherwise, the latent variable $H_i$ which does not have measured variables as children has $ne_T(H_i)= \emptyset$, which is always a subset of $ne_T(H_j)$, violating the assumption. In addition, note that Figures 1 in both [2] and [3] do not satisfy our definition of 'latent hierarchical structure'.
>
> Moreover, although the work in [1] can handle multi-level DAGs that some latent variables do not have measured variables as children, it requires that the underlying graph can be partitioned into multiple levels such that all the edges are between nodes in adjacent layers. In our work, we do not have such restrictions. For instance, Figure 1(b) in the main text and Figures 5-6 and Figures 7(a, b, d) in Appendix can be recovered by our method, but not by [1]. We have made it explicit in Appendix F.
>
> **Q2**: “Based on my understanding, the first and foremost assumption in this work is that the graph has a hierarchical structure. Note very importantly that this is a strict subset of directed acyclic graphs (DAGs). Therefore, this work does not handle general DAGs.”
>
> **A2**: Yes. It is a subset of DAGs. Compared to works [1,2,4,5] that handle continuous data, our assumptions on the structural constraints are much weaker. Furthermore, note that [3] assumes that the latent variables are **discrete**, where the discreteness assumption benefits the identification of more general DAGs among latent variables. In the next step, we will further make use of higher-order identifiability to achieve the identifiability of more general DAGs.
>
> **Q3**: “In the entire abstract and introduction, it is not mentioned that this paper is dealing with a linear model"
>
> **A3**: Thanks for raising this point. We have emphasized it in the abstract and introduction in the updated version. We are currently extending it to cover nonlinear relationships by leveraging kernels, with a similar search procedure.
>
> **Q4**: “Definition 2 is confusing. The phrase "Variables V are pure children of variables L" is defined. However, at this point, I don't see why $\mathbf{V}$ is uniquely defined, there could be multiple such Vs. Therefore the next sentence "we denote the pure children of L by $PCh_G(L)$" does not make sense"
>
> **A4**: That is a great catch. Thank you! We have updated Definition 2 as follows, where $V$ should be the **maximal** set that have no other parents than $\mathbf{L}$. Sorry for the confusion, and thanks to you--hope it is clear.
>
> Variables $\mathbf{V}$ are pure children of a set of latent variables $\mathbf{L}$ in a graph $G$, if $Pa(\mathbf{V}) = \mathbf{L}$ and $\mathbf{V} \cap \mathbf{L}=\emptyset$, and for any $\mathbf{V} \subset \mathbf{V}’$, $\mathbf{L} \subset Pa(\mathbf{V}’)$. That is, $\mathbf{V}$ is the maximal set that have no other parents than $\mathbf{L}$. We denote the pure children of $\mathbf{L}$ by $PCh(\mathbf{L})$.
>
> **Q5**: “More concretely, it's not clear how the Pure Children definition works if L has connections within itself. For example, take figure 1(a) and assume that the edge $L_1 \rightarrow L_3$ is not present for now. Suppose $L =$ {$L_1, L_3$}. Then $PCh_G(L) = $
> {$L_2, L_7, L_8, L_9$}”
>
> **A5**: Based on Definition 2, $PCh_G({L_1,L_3}) =$ {$L_2,L_4,L_5$}. Note that $L_7, L_8, L_9$ are not in $PCh_G(${$L_1,L_3$}$)$, because they have extra parents $L_2$ other than {$L_1,L_3$}.

---

### Official Review · Reviewer_eq5G · 2022-07-08

**Rating:** 4
**Confidence:** 3
**Soundness:** 3 good
**Presentation:** 4 excellent
**Contribution:** 3 good

**Summary:**

This paper proposes a new method to deal with causal discovery when latent confounds exist. The graph is assumed to be with hierarchical structure, and by incorporating the rank deficiency constraints, the method is able to learn the graph and some properties of the latent variables, which has been verified by experiments.


**Questions:**

1. How common do we meet the case that the causal structure is hierarchical？
2. It seems that def 2 is not neccessary.
3. I feel that even with phase 2, the rule 1 recovered graph may be incorrect since not all atomic covers are accessed. How to fix this problem?
4. From lemma 6, how to guarantee that the colliders are all identified?

**Limitations:**

No obvious limitations.

**Strengths And Weaknesses:**

Strengths:
1.	The paper is well formulated mathematically. Many definitions are clear and the theoretical analysis is sound.
2.	There are many examples to ease the understanding of the readers on the main concept of the paper.

Weaknesses:
1.	The experimental results are slightly weak. It seems only very simple toy data are evaluated.
2.	The efficiency of the method is not well studied. There are multiple phases of the method and the authors are better to show the computational aspects of the paper.

---

> ### Author Response · Authors · 2022-08-02
> **Responses to Reviewer eq5G**
>
> We appreciate your thoughtful comments and time devoted to reviewing this paper, and hope the following response properly addresses your concerns. We believe this work has a notable contribution to the causal discovery community.
>
> **Q1**: “The experimental results are slightly weak. It seems only very simple toy data are evaluated”
>
> **A1**: We performed various experiments on synthetic data to verify the behavior of the proposed approach; more experimental results were given in Appendix D, due to page limits.
>
> We are currently adapting the proposed method to very large-scale fMRI data analysis, together with domain experts to achieve appropriate interpretations, as an application of our approach to real, complex problems. There, we aim to automatically identify conceptually meaningful functional brain regions of different levels from the voxel data, where the lower level represents simpler functional regions and the higher level represents more abstract and complex functional regions.
>
>
> **Q2**: “The efficiency of the method is not well studied… better to show the computational aspects of the paper”
>
> **A2**: Thanks for the comments. The complexity of the algorithm is upper bounded by $O(r \sum_{k=2}^{l+1}\binom{n}{k})$, and this bound is further upper bounded by $O(r(1+n)^{l+1})$, where $n$ is the number of measured variables, and $l$ is the cardinality of the largest latent cover of the estimated graph, with $l \ll n$, and $r$ is the number of layers of the estimated hierarchical graph, with $r \ll n$. We have included this analysis on lines 57-60 in Appendix B in the revision.
>
> For instance, for the tested graph Figure 7(a) in our experiments, it takes around 40 seconds when the sample size is 2000, running on MacBook Pro M1.
>
> **Q3**: “How common do we meet the case that the causal structure is hierarchical”
>
> **A3**: The basic assumption in our hierarchical structure is that there is no loop; for a DAG, one can always organize the nodes according to the causal orders. Then the problem is whether the graphical assumptions in Condition 1 hold true for the DAG.  One can see that the latent hierarchical structure can explain a number of processes. For instance, for fMRI data analysis, hundreds of thousands of voxels are recorded, where these micro-variables may not necessarily have clear semantic meaning. Therefore, from the measured voxels, we aim to automatically identify conceptually meaningful functional brain regions of different levels, where the lower level represents simpler functional regions and the higher level represents more abstract and complex functional regions, which form a hierarchical structure. Similar hierarchical structures are also in the gene regulation process.
>
> Moreover, such hierarchical causal structure is also very common in image representation learning. With the input being measured pixels, we aim to extract hierarchical causal representations and concepts from images, which will be essential to image understanding and help to facilitate downstream tasks, such as prediction and decision-making.
>
> **Q4**: “It seems that def 2 is not necessary”
>
> **A4**: Definition 2 gives the definition of pure children. It is given for the reasons that it is crucial to structural identifiability, and this concept may not be familiar to general audiences, although it has been used in many works, such as [Spirtes et al., 2000; Silva et al., 2006; Xie et al., 2020].
>
> **Q5**: “I feel that even with phase 2, the rule 1 recovered graph may be incorrect since not all atomic covers are accessed”
>
> **A5**: We show in Theorem 12 in Appendix B that all latent atomic covers can be recovered with the first two phases (Algorithm 2 & Algorithm 3).  Please correct us in case there is any misunderstanding.
>
> **Q6**: “From lemma 6, how to guarantee that the colliders are all identified”
>
> **A6**: Lemma 6 guarantees the identifiability of each collider. Furthermore, with the search procedure given in Algorithm 4b, all colliders can be identified; due to the page limit, Algorithm 4b is given in Appendix A, but it is stated on lines 299-300 in the main text. Moreover, Theorem 7 gives the guarantee that the proposed algorithm can find the correct Markov equivalence class, which implies that all colliders can be correctly identified.

---

> > ### Author Response · Authors · 2022-08-06
> > **Could you kindly check whether our response and updated paper properly addressed your concern?**
> >
> > Dear Reviewer eq5G,
> >
> > Once again, we appreciate your time devoted to reviewing this paper.  We have provided responses to your comments and an updated submission. Could you please check whether they properly addressed your concern?  Your feedback would be appreciated.  Please kindly let us know in case there are other concerns--we hope we will have the opportunity to respond to them.  Thank you very much!

---

> > ### Author Response · Authors · 2022-08-09
> > **Could you please provide your feedback on our response?**
> >
> > Dear Reviewer eq5G,
> >
> > Thanks for providing the Author Rebuttal Acknowledgement. Given that the discussion involving authors will end in 6 hours, could you please let us know whether your main concerns were addressed by our response and updated submission?  If there is any other concern, please let us know, and we will immediately respond to it.
> >
> > If your main concerns are addressed, could you please update your recommendation to reflect it?  We understand you are very busy and appreciate your time.  Your feedback is valuable to us--we will be waiting for it.  Thank you.

---

### Official Review · Reviewer_548Y · 2022-07-13

**Rating:** 7
**Confidence:** 3
**Soundness:** 3 good
**Presentation:** 2 fair
**Contribution:** 3 good

**Summary:**

The authors present a method that under certain assumptions with respect to the graph structure, and an assumption of linear relationships between variables, guarantees identifiability of the Markov equivalent class of the hierarchical latent graph structure. The authors' method produce superior results to the nearest alternatives.

**Questions:**

Please see the section above for the questions presented to authors.

**Limitations:**

The authors briefly discuss the limitations of their work in the Conclusion section. Additional discussion might include the points raised above.

**Strengths And Weaknesses:**

The authors present a method that under certain assumptions with respect to the graph structure and form of the relationships between variables, guarantees identifiability up to MEC for latent hierarchical models - the method presented is potentially of interest to conference audience as it constitutes an improvement over prior methods.

What I was less clear on is in what scenarios would this method be useful for modeling observational data. When exactly would it be advantageous for a model selection to be conducted using this procedure? What would be the adventages compared to other methods for inferring latent structure. What is the extra adventage of having linear hierarchical latent variables? Or, do the authors consider this development as a potential foundation for other methods? Without these discussions, the method seems to be of interest to only a limited community of researchers.

How advantageous is the authors' method in terms of computational complexity compared to similar methods presented in the text? Are there any prohibitive trade-off's that come with the additional identifiability guarantees that the authors provide?

Lastly, I think the paper would benefit from improvements to its presentation. The authors present a lot of concepts (e.g. Defs 2,3,4) without any reference to their semantics with respect to the upcoming theoretical results. This leads the reader to do a lot of backtracking while examining the authors method.

---

> ### Author Response · Authors · 2022-08-02
> **Responses to Reviewer 548Y**
>
> Thanks a lot for your thoughtful comments and suggestions. Please see below for our responses to your specific comments.
>
> **Q1**: “in what scenarios would this method be useful for modeling observational data”
>
> **A1**: It is common to have hierarchical structures with only leaf nodes being measured in real-world scenarios, where our procedure should be applied. For instance, in fMRI data analysis, hundreds of thousands of voxels are recorded, where these micro-variables may not be necessary to have clear semantic meaning. Therefore, from the measured voxels, we aim to automatically identify conceptually meaningful functional brain regions of different levels, where the lower level represents simpler functional regions and the higher level represents more abstract and complex functional regions, which thus form a hierarchical structure.  We are currently working on large-scale fMRI data sets, together with domain experts to achieve appropriate interpretations, as an application of our approach to real, complex problems. We may also see similar structures in image representation learning--image pixels are dependent, and it seems sensible to consider them as observations generated by multiple-layer hidden concepts.
>
>
> **Q2**: “What would be the advantages compared to other methods for inferring latent structure”
>
> **A2**: The main advantage of the proposed method, compared to others that also handle latent variable graphs with continuous data (e.g., [Silva et al., 2006; Kummerfeld and Ramsey, 2016; Xie et al., 2020]), is that it relies on much weaker structural constraints, so that it can be applied to more general scenarios. Specifically, in our paper, (1) the pure children of the latent variable can be either latent or measured, i.e., allowing hierarchical structures,  (2) the number of pure children that each latent variable requires is much smaller, (3) each variable may have multiple latent parents, and moreover, (4) there can be multiple paths between every pair of variables, i.e., beyond the tree structure.
>
> **Q3**: “do the authors consider this development as a potential foundation for other methods?”
>
> **A3**: This is a great point! Yes, thanks for being thoughtful! We do consider this development as a potential foundation for other methods. This will be quite useful in scientific discovery, such as in neuroscience and biology. Moreover, we aim to extend it to allow nonlinear causal relationships with a similar search procedure, as well as allowing high-dimensional images as input. With such improvement, we can also extract hierarchical causal representations and concepts from unstructured data like images, which will be essential to understanding and help to facilitate downstream tasks, such as prediction and decision-making.
>
> **Q4**: “How advantageous is the authors' method in terms of computational complexity compared to similar methods presented in the text?”
>
> **A4**: In terms of computational complexity, our method is much faster than those that leverage higher-order statistics, such as GIN-based methods [Xie et al., 2020]. Other methods, such as [Silva et al., 2006, Kummerfeld and Ramsey, 2016], which also leverage rank tests, are more computationally efficient than ours, because they only handle much simpler graphs; for instance, FOFC [Kummerfeld and Ramsey, 2016] assumes that there is only one parent for each measured variable, so it only needs to test whether the rank $\leq 1$. For example, for Figure 7(a) with a sample size of 2000, our method takes 40 sec, while GIN [Xie et al., 2020] takes 95 sec.
>
> **Q5**: “The authors present a lot of concepts (e.g. Defs 2,3,4) without any reference to their semantics with respect to the upcoming theoretical results”
>
> **A5**: Thanks for the comments. Due to page limits, we put some explanations of those definitions in Appendix G.

---

> > ### Author Response · Authors · 2022-08-08
> > **Thank you again for your invaluable feedback**
> >
> > Dear Reviewer 548Y,
> >
> > Thank you once again for your time devoted to this paper and your invaluable feedback. We were wondering whether our response and updated paper have addressed your concern. Please kindly let us know in case there are other concerns. Many thanks!

---

> > ### Comment · Reviewer_548Y · 2022-08-09
> > **Thank you**
> >
> > I thank the authors for their response, and raise my score to reflect the concerns they addressed.

---

> > > ### Author Response · Authors · 2022-08-09
> > > **Thank you so much for checking the response and updating your recommendation**
> > >
> > > We are very happy that the response and updated paper are helpful to address your concerns. Your valuable comments have helped improve our presentation a lot. Thank you very much!

---

### Meta-Review · Area_Chair_xFjQ · 2022-08-28

**Recommendation:** Accept
**Confidence:** Certain

**Metareview:**

All reviewers and AC agree that this work is clearly of interest to NeurIPS.
Two out of three reviewers increased their score after their concrnes were successfully addressed
during the rebuttal. Reviewer's eq5G main concern was the weakness of the experimental results.
After reading the authors' response, the AC believes that the authors could introduce the additional supplementary
experiments they ran to cover the concern. A suggestion for better presentaiton is also communicated ot the authors.
Acceptance is recommended.

**Award:**

No

---

### Decision · Program_Chairs · 2022-09-14

Accept